# Nrf2 negatively regulates STING indicating a link between antiviral sensing and metabolic reprogramming

David Olagnier[1], Aske M. Brandtoft [1], Camilla Gunderstofte[1], Nikolaj L. Villadsen[2], Christian Krapp[1], Anne L. Thielke[1], Anders Laustsen[1], Suraj Peri[3], Anne Louise Hansen[1], Lene Bonefeld[1], Jacob Thyrsted[1], Victor Bruun[1], Marie B. Iversen[1], Lin Lin[1], Virginia M. Artegoitia [4], Chenhe Su[5], Long Yang[5], Rongtuan Lin[5], Siddharth Balachandran[3], Yonglun Luo [1,6], Mette Nyegaard [1], Bernadette Marrero[7], Raphaela Goldbach-Mansky[7], Mona Motwani[8], Dylan G. Ryan[9], Katherine A. Fitzgerald[8], Luke A. O'Neill[9], Anne K. Hollensen[10], Christian K. Damgaard[10], Frank v. de Paoli[1], Hanne C. Bertram[4], Martin R. Jakobsen [1], Thomas B. Poulsen[2] & Christian K. Holm[1]

The transcription factor Nrf2 is a critical regulator of inflammatory responses. If and how Nrf2 also affects cytosolic nucleic acid sensing is currently unknown. Here we identify Nrf2 as an important negative regulator of STING and suggest a link between metabolic reprogramming and antiviral cytosolic DNA sensing in human cells. Here, Nrf2 activation decreases STING expression and responsiveness to STING agonists while increasing susceptibility to infection with DNA viruses. Mechanistically, Nrf2 regulates STING expression by decreasing STING mRNA stability. Repression of STING by Nrf2 occurs in metabolically reprogrammed cells following TLR4/7 engagement, and is inducible by a cell-permeable derivative of the TCA-cycle-derived metabolite itaconate (4-octyl-itaconate, 4-OI). Additionally, engagement of this pathway by 4-OI or the Nrf2 inducer sulforaphane is sufficient to repress STING expression and type I IFN production in cells from patients with STING-dependent interferonopathies. We propose Nrf2 inducers as a future treatment option in STING-dependent inflammatory diseases.

[1] Department of Biomedicine, Aarhus Research Center for Innate Immunology, Aarhus University, Bartholin Alle 6, 8000 Aarhus, Denmark. [2] Department of Chemistry, Aarhus University, Langelandsgade 140, 8000 Aarhus, Denmark. [3] Fox Chase Cancer Center, 333 Cottman Avenue, Philiadelphia, PA 19111-2497, USA. [4] Department of Food Science, Aarhus University, Kirstinebjergvej 10, 5792 Aarslev, Denmark. [5] Lady Davis Institute-Jewish General Hospital, McGill University, Division of Experimental Medicine, 3755 Côte Ste-Catherine Road, Montreal, QC H3T 1E2, Canada. [6] BGI-Shenzhen, Beishan Industrial Zone, Yantian District, Shenzhen 518083, China. [7] Translational Autoinflammatory Diseases Studies, NIAID, NIH, 10 Center Drive, Bethesda, MD 20892, USA. [8] Program in Innate Immunity, Division of Infectious Diseases and Immunology, Department of Medicine, University of Massachusetts Medical School, 364 Plantation Street, Worcester, MA 01605, USA. [9] School of biochemistry and Immunology, Trinity Biomedical Sciences Institute, Trinity College, College Green, Dublin 2 D02 PN40 Ireland, UK. [10] Department of Molecular Biology and Genetics, Aarhus University, C.F. Moellers Allé 3, 8000 Aarhus, Denmark. These authors contributed equally: Aske M. Brandtoft, Camilla Gunderstofte. Correspondence and requests for materials should be addressed to D.O. (email: olagnier@biomed.au.dk) or to C.K.H. (email: holm@biomed.au.dk)

Nrf2 (Nuclear factor (erythroid-derived 2) -like 2) is a member of the cap´n´collar basic leucine zipper family of transcription factors characterized structurally by the presence of Nrf2-ECH homology domains[1]. At steady state, Nrf2 is kept inactive in the cytosol by its inhibitor protein Keap1 (Kelch-like ECH-associated protein 1), which targets Nrf2 for proteasomal degradation[2]. In response to oxidative stress, Keap1 is inactivated and Nrf2 is released to induce the transcription of Nrf2-responsive genes. In general, the genes under the control of Nrf2 protect against stress-induced cell death and Nrf2 has thus been suggested as the master regulator of tissue damage during infection[3]. Furthermore, Nrf2 is also an important regulator of the inflammatory response[4,5] and was recently identified to function as a transcriptional repressor of inflammatory genes in murine macrophages[6].

Type I IFNs (IFNα and -β) are central to immune-protection against infection with virus. Production of IFNα/β in response to infection is highly dependent on innate recognition of cytosolic viral nucleic acids by cellular pathogen recognition receptors (PRRs). These receptors include the RNA sensors RIG-I and MDA-5, which signal through the adaptor MAVS[7,8], and the DNA sensor cGAS which signals through the adaptor STING[9–12]. In both signaling pathways, binding of viral nucleic acids to their respective sensors leads to recruitment and phosphorylation of the kinase TBK1 (Tank Binding Kinase 1), which in turn activates the IRF3 transcription factor by phosphorylation[13–15]. Whereas a balanced production of type I IFNs is necessary for protection against virus, excessive production hereof is a powerful driver of pathology. This has recently been demonstrated in influenza A virus infections[16] as well as in a series of auto-inflammatory disorders such as systemic lupus erythematosus[17,18] and in the more recently discovered disease STING-associated vasculopathy with onset in infancy (SAVI)[19]. In the latter case, gain-of-function mutations in STING drives a systemic and debilitating inflammatory condition[19]. Tight regulation of type I IFNs is thus necessary to avoid excessive immune mediated tissue damage in infection as well as in homeostasis. If and how Nrf2 affects type I IFN responses induced by antiviral cytosolic sensing and if the Nrf2/Keap1 axis is a potential target for treating STING-dependent interferonopathies is, however, not currently known.

The role of biochemistry has recently gained a newfound foothold in innate immunology. Studies dating back from the 1970s showed that microbial products, such as LPS (lipopolysaccharide), negatively regulate respiration of macrophages by inhibiting complexes in oxidative phosphorylation[20,21]. These early discoveries have now formed the basis of a completely new area of immunology referred to as immunometabolism[22]. Metabolic reprogramming is now known to include an increase in glycolysis and a two-point interruption of the tricarboxylic acid (TCA) cycle[23,24]. Recent work has now demonstrated that an important result of metabolic reprogramming, induced through stimulation with LPS, is the accumulation of distinct TCA-cycle derived metabolites—in particular succinate and itaconate[25,26]. Earlier work demonstrated that succinate operates as a pro-inflammatory agent and is important for the release of IL-1β[25]. The anti-inflammatory effect of endogenous itaconate was initially described in Irg1 deficient murine macrophages that lack itaconate production[27]. Further, itaconate has been demonstrated to have anti-inflammatory properties by inhibiting the enzymatic activity of succinate dehydrogenase (SDH) to accumulate succinate[25–27]. Moreover, a recent report demonstrated that a cell-permeable derivative of itaconate (4-octyl-itaconate, 4-OI) blunts transcription of IL-1β through activation of the transcription factor Nrf2, which acts as a repressor of IL-1β transcription[28]. Altogether, these reports contribute to a growing body of evidence for a dependency on metabolic reprogramming for the control of pro-inflammatory cytokine release. No reports have so far demonstrated a link between cellular accumulation of metabolites and regulation of antiviral cytosolic sensing.

In this study, we demonstrate that Nrf2 represses antiviral cytosolic sensing by suppressing the expression of the adaptor protein STING. Further, we show that the repression of STING expression occurs during metabolic reprogramming following TLR ligation and is inducible by 4-OI, the cell-permeable derivative of the TCA-cycle metabolite itaconate, through activation of Nrf2. Finally, treatment with Nrf2 inducers, including 4-OI, sufficiently reduces STING-dependent release of type I IFNs from SAVI-derived fibroblasts promoting the idea that Nrf2 is a valid target in STING-associated inflammatory disorders.

## Results

**Nrf2 represses STING mRNA and protein levels in human cells.** To test whether the transcription factor Nrf2 affects the innate antiviral program, we used a full transcriptome analysis to identify differentially regulated pathways and genes in human cells silenced for Nrf2. As an experimental tool, we used the human epithelial cell line A549 which harbors a loss-of-function mutation in Keap1, thus rendering Nrf2 constitutively active in these cells[29]. A total of 1511 genes were significantly regulated between control and Nrf2 silenced cells (Supplementary Fig. 1a). A functional clustering (node analysis) using this gene list identified an intersection between Nrf2 signaling with antiviral-associated pathways (Fig. 1a). Among the differentially expressed genes were several genes involved in type I (α/β) interferon response and in antiviral sensing and signaling—including TMEM173, the gene encoding the innate adaptor molecule STING[30] (STimulator of INterferon Genes) (Fig. 1b, c and Supplementary Fig. 1b–c). STING is central to the release of type I IFNs in response to cytosolic DNA and signals downstream of the cytosolic DNA sensor cGAS[9,31,32]—suggesting that Nrf2 is a negative regulator of STING-dependent type I IFN responses. In contrast, transcripts from known Nrf2-inducible genes such as Nrf2 itself (NFE2L2), HMOX-1[33], and NQO1[34] were all decreased by siRNA-mediated silencing of Nrf2 (Fig. 1b and Supplementary Fig. 1b). We were able to validate the regulatory role of Nrf2 on TMEM173 mRNA levels by multiple qPCR experiments in A549 and also in human keratinocyte HaCat cells, which were instead silenced for the Nrf2 repressor Keap1 (Fig. 1d–g). The regulation of TMEM173 mRNA was sufficient to affect STING protein levels as A549 cells lacking Nrf2, following CRISPR/Cas9 gene editing or treatment with two different Nrf2 targeting siRNAs, displayed markedly increased levels of STING protein while other major signaling molecules such as cGAS, IFI16, RIG-I, MDA5, MyD88, and TBK1 were unaffected (Fig. 1h and Supplementary Fig. 2a). In line, both CRISPR/Cas9 silencing of Nrf2 in A549 and siRNA-mediated Keap1 silencing in HaCat cells affected STING expression—again without affecting other important signaling molecules (Supplementary Fig. 2a and Fig. 1i). Further, treatment of A549 cells with the Nrf2 inhibitor ML385[35] led to decreased expression of HO-1 and increased expression of STING (Supplementary Fig. 2c). The effect of the Nrf2/Keap1 pathway on STING protein expression was quantified and statistically confirmed in multiple western blot experiments (Fig. 1j, k). Notably, the repression of STING by Nrf2 was restricted to human cells as the lack of Nrf2 in murine bone-marrow derived macrophages from Nrf2$^{-/-}$ mice or Nrf2/Keap1 CRISPRed RAW 264.7 macrophages did not interfere with STING expression (Supplementary Fig. 3). Importantly, silencing of Nrf2 in primary human monocyte-derived macrophages (hMDMs) also led to increased STING mRNA and protein levels demonstrating that the regulatory effect of the Nrf2/Keap1 axis on STING was not limited to

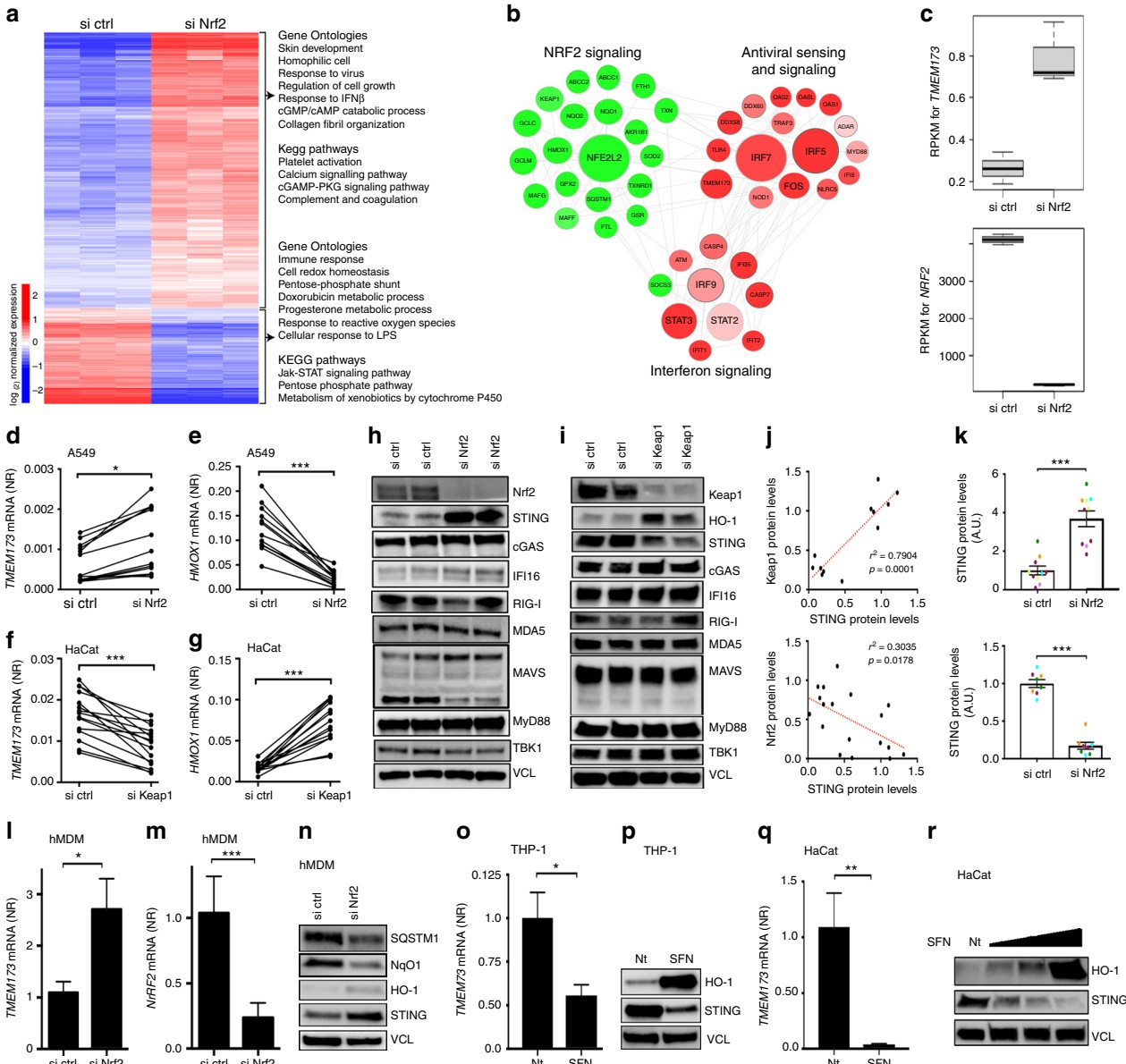

**Fig. 1** Nrf2 suppresses STING expression. A549 cells were transfected with control or Nrf2 siRNA for 48 h. Samples were analysed by RNAseq and differentially expressed pathways and genes that satisfied p-value <0.001 and fold change cutoff >1 or <−1 were selected. The data are from one experiment performed in triplicate. **a** Heat map of differentially regulated genes and a listing of representative gene ontologies and KEGG pathways associated with these genes. **b** Cloud analysis of differentially expressed genes. **c** Representation of RNA-reads for from RNAseq. Graph displays boxplots with boxes indicating mean, s.e.m. as well as min and max values. **d–g** Cells were transfected with control, Nrf2 or Keap1 siRNA for 48 h or 72 h, respectively. *TMEM173* and *HMOX-1* mRNA expression levels were monitored by qPCR. Each two connected dots represent data from one biological experiment with paired samples. **h**, **i** Antiviral sensor, adaptor or accessory molecules were examined by immunoblotting. Data are representative of two independent experiments. **j** Correlation between Keap1 and STING protein levels in HaCat cells and Nrf2 and STING protein levels in A549 cells using Image J quantification of western blot intensities. Correlations were calculated using a Spearman's test. Each dot represents a biological sample. **k** Quantification of STING staining intensities in multiple western blots using Image J. Colorations indicate sample pairing. Graphs display mean and s.e.m. **l–n** *TMEM173* and *NRF2* mRNA expression levels by qPCR (**l**, **m**) or western blotting (**n**) in human monocyte-derived macrophages treated with control siRNA or Nrf2 siRNA. Data represent means ± s.e.m. of experiments performed on two donors. **o–r** PMA-differentiated THP1 cells (**o**) or HaCat cells (**q** and **r**) were stimulated with ʟ-sulforasphane (SFN) (20 μM) for 72 h. *TMEM173* mRNA and STING protein levels were determined by qPCR and immunoblotting, respectively. Data are from one representative experiment which has been repeated twice. Two-tailed Student's t-test was used to determine statistical significance

cell lines but also occurred in primary human cells (Fig. 1l–n). To approach the role of Nrf2 in a complimentary manner, we demonstrated that treatment of HaCat cells and cells of the human monocytic lineage THP1 with the chemical Nrf2 activator sulforaphane (SFN) (Supplementary Fig. 4) led to decreased STING mRNA and protein levels (Fig. 1o, p). Further, using

HEK293 cells we investigated the effect of STING expression when Nrf2 and STING were co-expressed. Here, co-expression of Nrf2 greatly reduced expression of STING (Supplementary Fig. 5a). By contrast, we were unable to affect STING expression by silencing of the autophagy regulatory proteins ULK1[36] (Unc-51 like autophagy activating kinase 1), ATG7 (Autophagy-related

protein 7)[37], or SQSTM1(p62)[38] in A549 cells (Supplementary Fig. 6). Further, co-silencing of these autophagy proteins did not affect the suppression of STING expression in HaCat cells by removal of Keap1, thus demonstrating that suppression of STING is autophagy-independent in this context (Supplementary Fig. 6). Impairment of STING expression by Nrf2 also seemed independent of type I IFN and of increased reactive oxygen species (ROS) accumulation as neither silencing of IFNAR nor treatment with the ROS scavenger L-NAC affected STING suppression. Furthermore, no induction of *IFNβ1* mRNA was observed by Nrf2 silencing (Supplementary Fig. 7, 8). Additionally, individual silencing of a cross-section of Nrf2 differentially regulated genes uncovered from our RNAseq analysis did not modulate STING expression in A549 cells (Supplementary Fig. 8). No physical interaction between Nrf2 and STING protein was detected as we were unable to co-precipitate Nrf2 with STING in immunoprecipitation experiments using HEK293 where both STING and Nrf2 were overexpressed (Supplementary Fig. 9). Finally, treatment of A549 with the proteasomal inhibitor MG-132 did not significantly elevate STING protein levels while the protein levels of MDM2 and cyclin B, which are known to be sensitive to MG-132, were highly affected by the treatment (Supplementary Fig. 9).

**Nrf2 regulates STING mRNA stability**. A recent report demonstrated that Nrf2 can bind to a DNA region in the proximity of the IL-1β promoter in murine macrophages[6]. From this position, Nrf2 decreases the recruitment of the RNA polymerase II (Pol II) to the IL-1β gene and hereby negatively regulates IL-1β expression[6]. We hypothesized that Nrf2 regulates STING expression in a similar manner in human cells and thus performed a ChIP-seq analysis using a Nrf2 specific antibody and A549 cells in which Nrf2 is constitutively active. Here we found that although Nrf2 peaks were readily detectable in the proximity of known Nrf2 regulated genes such as *NQO1*, no detectable Nrf2 peak was present in proximity to *TMEM173* locus (Fig. 2a, b). To decrease the risk of having erroneously disregarded or overlooked a Nrf2 peak we also performed the peak calling with a more relaxed cutoff (*p*-value 1e−3). With these looser settings, we identified a small Nrf2 peak located ~25 kb from *TMEM173* (Supplementary Fig. 10a). However, deletion of this genomic region in A549 cells with CrisprCas9 technology led to a slight decrease rather than an increase in STING expression (Supplementary Fig. 10b) thereby supporting the finding that Nrf2 does not regulate STING through a direct genomic interaction in a region in close proximity to *TMEM173*. As the effect of Nrf2 on *TMEM173* transcription might be indirect we then performed another ChIP-seq, this time using an RNA Pol II-specific antibody and A549 cells treated with either control or Nrf2 targeted siRNA. Much to our surprise, presence of RNA Pol II on *TMEM173* was below the level of detection in both of the siRNA-treated groups. By sharp contrast, recruitment of the RNA Pol II to the Nrf2-regulated gene *TXNRD1* was detectable in both groups and was highly impaired by the silencing of Nrf2 (Fig. 2c, d). The relatively limited presence of RNA Pol II in the *TMEM173* gene suggested a weak transcriptional control of *TMEM173* in A549 cells and evoked the possibility of a post-transcriptional regulation of *TMEM173* gene by Nrf2. This was also supported by the relatively low number of reads for *TMEM173* detected by RNAseq (Fig. 1c). As we had already determined that Nrf2 affected *TMEM173* mRNA levels, we hypothesized that Nrf2 regulates *TMEM173* mRNA levels by affecting mRNA stability. To test this hypothesis, we used A549 cells treated with either control or Nrf2 targeted siRNA and further treated with actinomycin D (ActD). ActD binds to DNA at the transcription initiation complex and prevents elongation of

RNA chain by RNA polymerase, thus inhibiting de novo transcription. We then determined how *TMEM173* and *MB21D1* (cGAS) mRNA levels changed over time after transcription was blocked. Interestingly, whereas *MB21D1* mRNA levels quickly decreased, *TMEM173* mRNA levels were stable within the first 4 h of ActD exposure and there was no observable effect of Nrf2 silencing (Fig. 2e, f). These data demonstrated that *TMEM173* mRNA is highly stable and that STING and cGAS expression is likely to be regulated in different manners. We then expanded on these experiments to include time points beyond 4 h. Here we were able to observe a significant decrease in *TMEM173* mRNA levels in the cells treated with control siRNA exposed to ActD from 6–18 h. Interestingly, in the Nrf2 siRNA-treated cells *TMEM173* mRNA levels remained stable and hereby demonstrated that Nrf2 affects STING mRNA stability and suggests that this is the mechanism by which Nrf2 controls STING expression (Fig. 2g). Notably, the stability of *MB21D1* mRNA was unaffected by Nrf2 silencing at these time points and the stability of Nrf2 mRNA seemed to be even decreased when Nrf2 was silenced (Fig. 2h, i). The same findings were observed in HaCat keratinocytes that were silenced for Keap1 (Fig. 2j). On a prolonged period of ActD, stability of *TMEM173* mRNA was lowered by the lack of Keap1 (Fig. 2k). Importantly, the stability of cGAS mRNA was unaffected following Keap1 silencing (Fig. 2l). As mRNA stability and efficiency of translation is often interdependent, we performed a polysome profiling of *TMEM173* mRNA in HaCat cells treated with either control or Keap1 siRNA. Here, the association of STING mRNA with polysomes was unaltered between the two treatment groups and thus we suggest that changes in translation efficiency is not a strong regulator of STING expression by the Nrf2/Keap1 pathway (Supplementary Fig. 11). Together these data demonstrate that Nrf2 regulates STING mRNA stability and suggest that this process is a post-transcriptional regulatory mechanism by which Nrf2 can control STING expression in human cells.

**Nrf2 regulates STING-dependent antiviral immune responses**. As Nrf2 repressed STING expression, we further asked whether impairment of STING expression was sufficient to affect its antiviral functionality and capacity to respond to cytosolic DNA and cGAMP. Indeed, silencing of Nrf2 in A549 and of Keap1 in HaCat cells affected downstream STING signaling events including phosphorylation of TBK1 and IRF3[13] in response to the STING agonist cGAMP (Fig. 3a, b). Additionally, expression of *IFNβ1* and subsequent induction of IFN-stimulated genes were greatly affected by silencing of Nrf2 either by siRNA or by CRISPR/Cas9 editing (Fig. 3c, d). This was also the case when Nrf2 was activated in THP1 cells by pre-treatment with SFN prior to engagement with cGAMP (Fig. 3e–g). As STING is critical for innate protection to infection with DNA viruses in both of these cell types (A549 and HaCat) (Supplementary Fig. 12), we tested the effect of removing Nrf2 on the susceptibility to herpes simplex virus (HSV) infection. Here, silencing of Nrf2 in A549 cells decreased infectivity with a GFP-expressing HSV-1 strain (KOS) (Fig. 3h) and suppressed the release of progeny virus when infected with two different HSV-2 strains (MS and 333) (Fig. 3i, j). In further support of Nrf2 regulating a STING-dependent antiviral response we observed that in HEK293 cells stably expressing STING the expression of Nrf2 reduced the IFN-response to known STING agonists, including HSV-1 (Supplementary Fig. 5b–d).

**Nrf2 suppresses STING during metabolic reprogramming**. Metabolic reprogramming in immune cells is inducible in macrophages or dendritic cells by stimulation with LPS[25,39]. LPS is

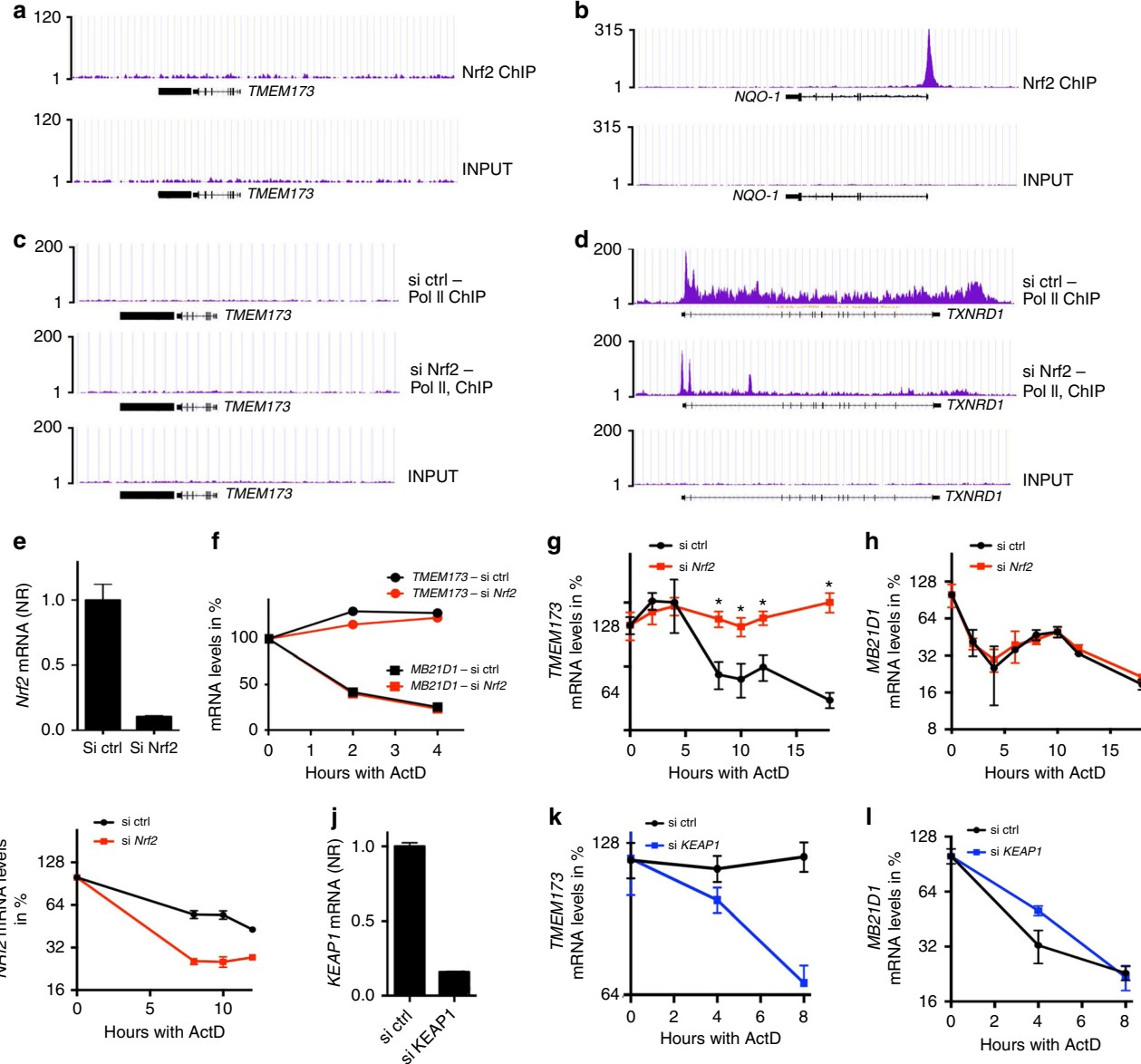

**Fig. 2** Nrf2 regulates STING mRNA stability. **a**, **b** A549 cells were lysed and subjected to ChIP-seq using a Nrf2 specific antibody. Graphs display fragment pileup per 1 million reads at the genomic loci proximal to *TMEM173* (**a**) and *NQO-1* (**b**). **c**, **d** A549 cells were treated with either control (ctrl) or Nrf2 targeted siRNA for 48 h. Cells were then lysed and subjected to ChIP-seq analysis using a Pol II-specific antibody. Graphs display fragment pileup per 1 million reads at the genomic loci surrounding *TMEM173* (**c**) and *TXNRD1* (**d**). **e–i** A549 cells were treated with ctrl or Nrf2 siRNA for 48 h. Knockdown efficiency of Nrf2 was determined by qPCR (**e**). **f–i** Cells were then treated with Actinomycin D (10 μg/mL) as indicated and gene expression was determined by qPCR. The graph in (**f**) is one representative of two independent experiments. The graphs in (**g**) and **h** contains merged data from two independent experiments with $n = 5$ at the 8 h timepoint and $n > 2$ for remaining time points. Data displayed in (**i**) represent one experiment with $n = 3$. (*) indicates statistical significance using Student's *t*-test with $p < 0.05$. **j–l** HaCat cells were treated with ctrl or Keap1 siRNA for 72 h. Knockdown efficiency of Keap1 was determined by qPCR (**j**). **k**, **l** Cells were then treated with Actinomycin D (5 μg/mL) as indicated and gene expression was determined by qPCR. Data displayed in (**k**) and **l** represent one experiment with $n = 3$. (*) indicates statistical significance using Student's *t*-test with $p < 0.05$

long known to activate Nrf2[40]. Thus, we hypothesized that Nrf2 could also be a repressor of STING in a context of metabolic reprogramming. As metabolic reprogramming is triggered by toll-like receptors (TLRs) we chose a human pDC cell line (PMDC05), which is known to be highly sensitive to stimulation through these receptors[41,42]. As expected, we were able to demonstrate that treatment of PMDC05 cells with the TLR4 agonist LPS or the TLR7 agonist gardiquimod led to increased release of lactate, increased glucose consumption, enhanced accumulation of itaconate, as well as increased expression of HIF1α—the hallmarks of metabolic reprograming in innate

immune cells (Fig. 4a–c). Notably, in this context we were able to observe a time-dependent repression of STING expression in a manner very similar to that observed with a genetic or chemical activation of Nrf2 (Fig. 4e, f). By contrast, expression of other innate signaling molecules such as RIG-I, MAVS, and MDA-5 was not reduced but rather increased in response to TLR ligation (Fig. 4e). Notably, the expression of the Nrf2 inducible enzyme heme-oxygenase-1 (HO-1) was enhanced following TLR4 or TLR7 agonist treatment supporting a possible link between Nrf2 activation and STING suppression during LPS-induced metabolic reprogramming. This link was confirmed as treatment of

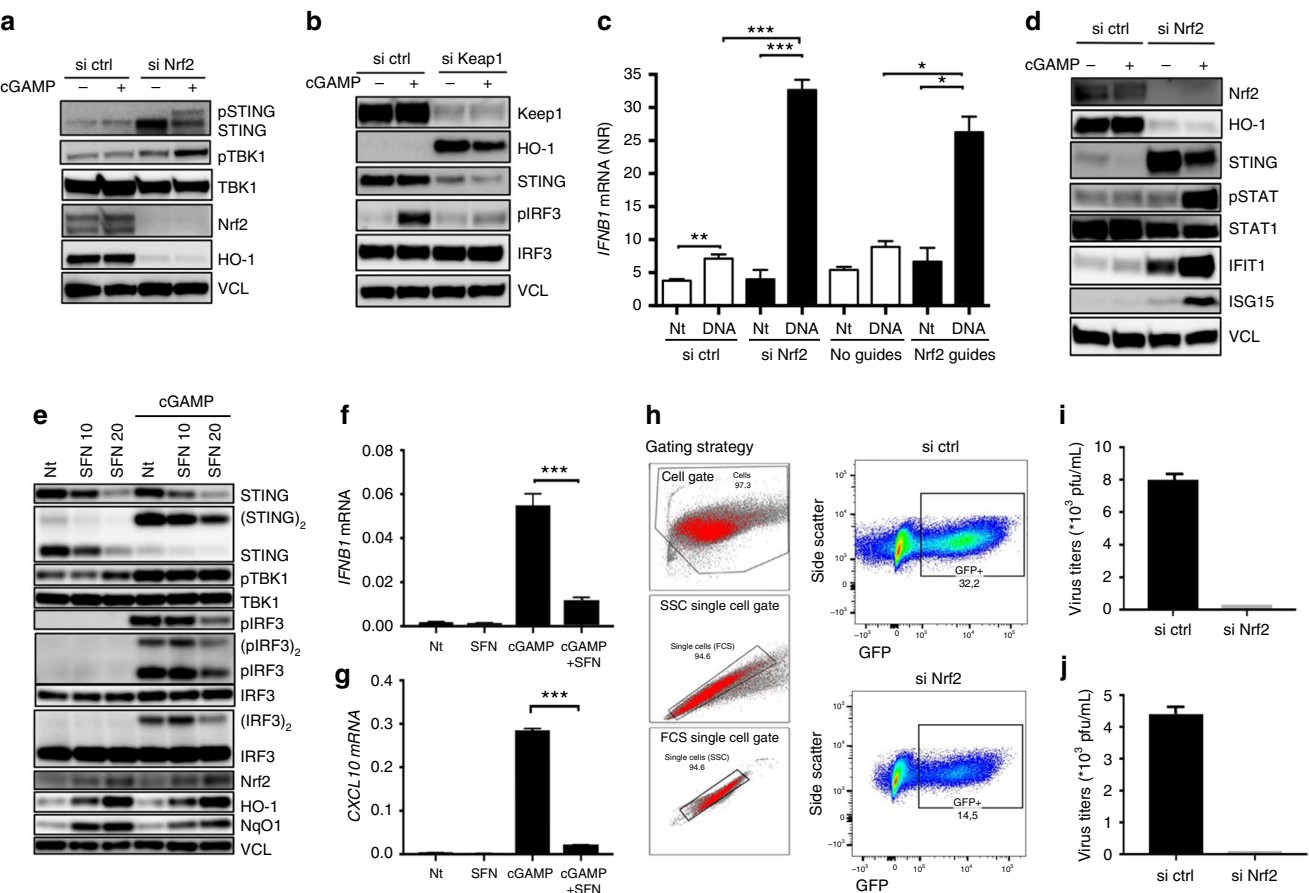

**Fig. 3** Nrf2 impairs antiviral innate immunity and increases susceptibility to DNA virus infection. **a**, **b** A549 cells (**a**) or HaCat cells (**b**) were transfected with Nrf2 or Keap1 siRNA for 48 h or 72 h, respectively, and were subsequently challenged with cGAMP for 3 h (4 µg mL$^{-1}$). Whole-cell extracts were analysed for STING signaling events by immunoblotting. Blots in (**a**) and **b** are representative of >6 independent experiments. **c** A549 cells silenced for Nrf2 using siRNA or lacking Nrf2 after CRISPR/Cas9 editing were stimulated with dsDNA (4 µg mL$^{-1}$). *IFNB1* mRNA levels were determined by qPCR. Data are the means ± s.e.m. of an experiment performed in triplicate. **d** A549 cells were transfected with control or Nrf2 siRNA for 48 h and were subsequently challenged with cGAMP for 24 h (4 µg mL$^{-1}$). Whole-cell extracts were analysed for ISG expression by immunoblotting. Blot is representative of more than four independent experiments. **e–g** PMA-differentiated THP1 cells were pre-treated with increasing doses of L-sulforaphane (SFN) for 72 h and subsequently stimulated with cGAMP (4 µg mL$^{-1}$) for 3 h (**f**) or 6 h (**f**, **g**). Whole-cell extracts were analysed for antiviral signaling events by immunoblotting (**e**) and *IFNβ1* and *CXCL10* gene expression by qPCR (**f**, **g**). Blot in **e** is representative of three independent experiments. Data in (**f**) and **g** are the means ± s.e.m. of an experiment performed in triplicate. **h–j** A549 cells were transfected with control or Nrf2 siRNA for 48 h and were subsequently infected with HSV-1-GFP (KOS strain), or HSV-2 (MS and 333 strains) at an MOI of 0.01 for 24 h. Viral infectivity was determined by flow cytometry (**h**) and viral replication by plaque assay (**i**, **j**). **h** Graphs display gating strategy (left) and GFP intensity by flow cytometry (right). **i**, **j** Graphs display mean and s.e.m. of plaques two independent experiments. Data presented in (**h–j**) are representative of two independent experiments. Unpaired two-tailed Student's *t*-test was used to determine significance of the difference between the control and each experimental condition

PMDC05 cells with the Nrf2 inhibitor ML385 largely abolished the effect of LPS and gardiquimod on STING expression (Fig. 4g). The suppression of STING expression by TLR ligation was sufficient to significantly inhibit its functionality as pre-treatment with LPS abolished the release of type I IFNs in response to the STING agonist cGAMP (Fig. 4h). A recent report demonstrated the electrophilic nature of the citrate-derived metabolite itaconate and its ability to activate the Nrf2 pathway through direct binding of Keap1[43]. At the same time, the O'Neill's group demonstrated that also the itaconate derivative 4-OI can bind Keap1 to promote Nrf2 activity[28]. We therefore hypothesized that 4-OI might also be able to affect STING expression through activation of Nrf2 (Fig. 4i). We were able to confirm 4-OI as a potent Nrf2 activator as we observed a dose-dependent increase in a luciferase-based Nrf2-promoter activity assay in response to 4-OI (Fig. 4j). Then, a strong reduction in *TMEM173* mRNA levels was observed following stimulation with 4-OI in cells of the human monocytic cell line THP1, thus supporting the existence of an itaconate/Nrf2

link in the regulation of STING. By contrast, 4-OI did not affect cGAS (*MB21D1*) expression levels and expression of the Nrf2-inducible gene *HMOX-1* was highly increased (Fig. 4k). Further, treatment with 4-OI was sufficient to greatly impair STING protein expression and STING-dependent signaling in response to cGAMP (Fig. 4l). The strength of this suppression was sufficient to also inhibit release of type I IFNs as well as downstream STAT1 signaling and subsequent induction of ISGs (Fig. 4m, n). The effect of 4-OI treatment depended on Nrf2 as silencing hereof by siRNA decreased the suppression of STING in HaCat cells (Fig. 4o and Supplementary Fig. 14) and in A549 cells (Fig. 4p). We also performed siRNA-mediated silencing of Nrf2 in primary hMDMs. Here, silencing of Nrf2 largely abolished the effect of 4-OI on STING expression in cells from two different healthy human donors (Fig. 4q–s and Supplementary Fig. 13). These data demonstrate that STING expression is repressed by LPS and by 4-OI through activation of Nrf2 and hereby suggest

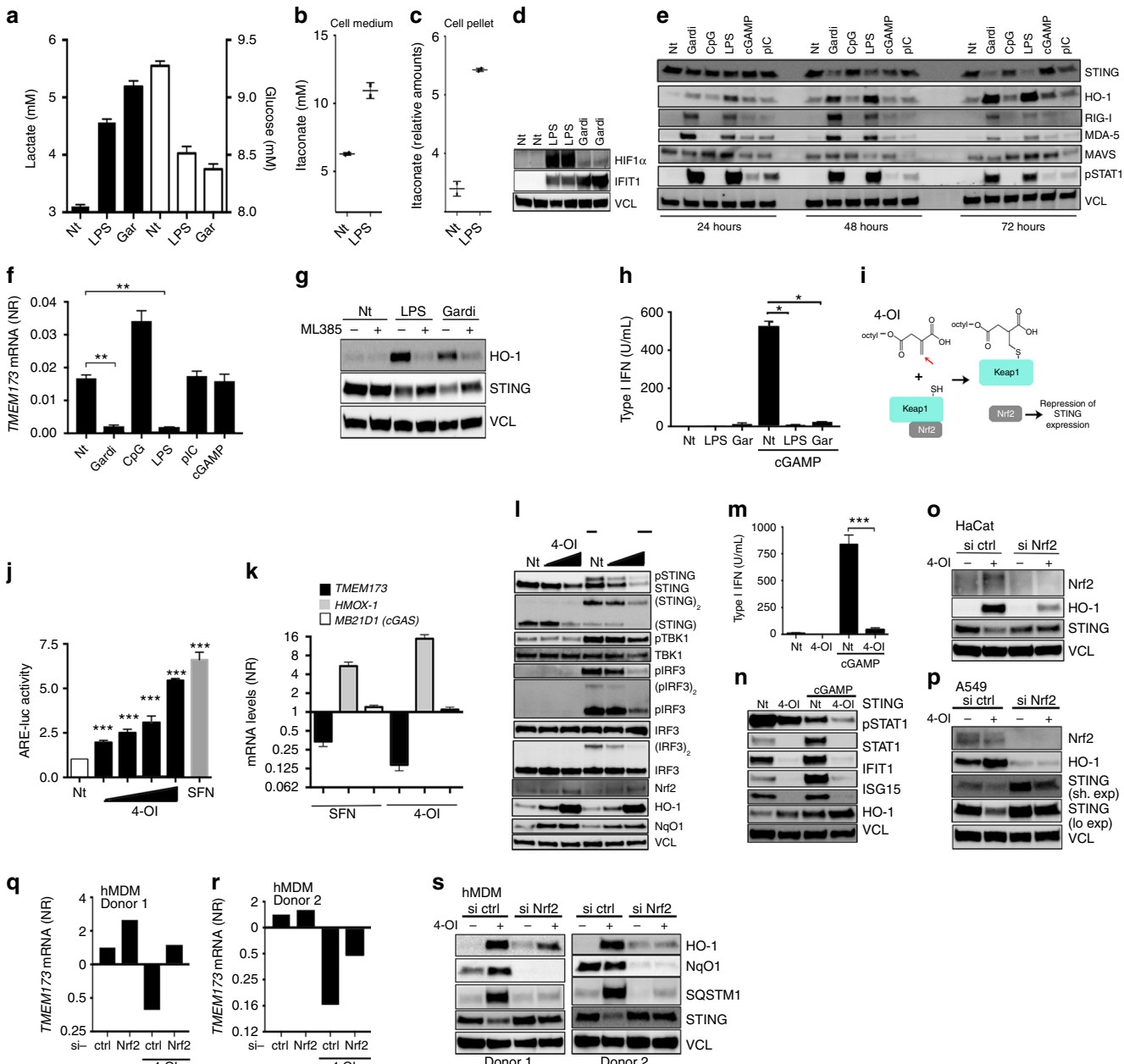

**Fig. 4** Nrf2 is a negative regulator of STING during metabolic reprogramming. **a** Lactate and glucose were measured in supernatants from PMDC05 cells stimulated with LPS (1 µg mL$^{-1}$) or gardiquimod (4 µg ml$^{-1}$) for 24 h. Data are the means and s.e.m. from one of two independent experiments performed in septuplicates. **b**, **c** Itaconate was measured in the cell medium and in the cell pellets of LPS-stimulated PMDC05 by LC/MS. Data are the means and s.e.m. from one of two independent. **d** PMDC05 stimulated with LPS (200 ng mL$^{-1}$) or gardiquimod (400 ng ml$^{-1}$) for 24 h were lysed and analysed for HIF1α, IFIT1, and Vinculin (VCL) as loading control. **e** PMDC05 cells were stimulated with gardiquimod (4 µg mL$^{-1}$), LPS (1 µg mL$^{-1}$), CpG (4 µg mL$^{-1}$), cGAMP (4 µg mL$^{-1}$), or Poly (I:C) (4 µg mL$^{-1}$) for 24, 48, or 72 h. Whole-cell extracts were analysed by immunoblotting. **f** TMEM173 mRNA was assessed by qPCR in PMDC05 stimulated with indicated agonists for 48 h. Data are the means ± s.e.m. of one experiment performed in triplicate. **g** PMDC05 cells were pre-treated with the Nrf2 inhibitor ML385 (20 µM) before stimulation with LPS (1 µg mL$^{-1}$) or gardiquimod (4 µg mL$^{-1}$) for 72 h. Cells were then lysed and lysates were analysed by western blotting. **h** PMDC05 cells were pre-stimulated with gardiquimod (4 µg mL$^{-1}$) or LPS (1 µg mL$^{-1}$) for 72 h before challenge with cGAMP (4 µg mL$^{-1}$) for 24 h. Type I IFN release was assessed using a HEK-Blue assay. Data are the means ± s.e.m. of one representative experiment performed in triplicate. **i** Graphical display of how 4-OI might possibly activate Nrf2. **j** HEK293T cells were treated with increasing doses of 4-OI (30–250 µM) for 18 h. ARE-promoter activity was assessed by luciferase assay. Data are means ± s.e.m. of two independent experiments in triplicate. **k** THP1 cells were treated with SFN (20 µM) or 4-OI (125 µM) for 48 h and mRNA levels were determined by qPCR. **l** THP1 cells were pre-treated with increasing does of 4-OI (62.5–125 µM) for 72 h and challenged with cGAMP (4 µg mL$^{-1}$). Whole-cell lysates were then blotted as indicated. Data are from one representative experiment that has been repeated twice. **m–n** THP1 cells were pre-treated with 4-OI (125 µM) for 72 h before challenge of 24 h with cGAMP (4 µg mL$^{-1}$). Supernatants were assessed for type I IFN by HEK-Blue cell assay (**m**) and whole-cell lysates by immunoblotting (**n**). **o**, **p** HaCat and A549 cells were treated with Nrf2 siRNA for 72 h and for 48 h, respectively, before treatment with 4-OI (125 µM) for 48 h. Whole-cell lysates were used for immunoblotting. **q–s** mRNA levels (**q**, **r**) and protein levels (**s**) were assessed by qPCR and immunoblotting in hMDMs silenced for Nrf2 by siRNA. Data are from two donors. Unpaired two-tailed Student's t-test was used to determine significance

that metabolic reprogramming can control cytosolic antiviral DNA sensing.

**4-OI and SFN repress STING and IFN in SAVI fibroblasts.** In addition to its protective function during infection, STING can also be a strong inducer of immunopathology. This has recently been demonstrated in the so-called SAVI patients[19] who harbor gain-of-function point mutations in the exon 5 of STING gene (V174L, N152S, or V155M)[19]. Such STING variants are able to secrete type I IFN spontaneously and activate an IFNβ Luciferase reporter construct without exogenous addition of STING sti-muli[19]. First, we evaluated whether induction of Nrf2 using SFN or 4-OI could reduce STING mRNA levels in HEK293T cells transiently expressing WT-STING or SAVI-STINGs (V174L, N152S, or V155M). Interestingly, treatment of the STING expressing cells with SFN or 4-OI reduced by more than twofold *TMEM173* mRNA levels (Fig. 5a, b). We then examined whether the Nrf2 inducers could also reduce the activity of the different SAVI-STINGs. Transient expression of the SAVI-STINGs resul-ted in high activation of the ISRE promoter, phosphorylation of STAT1, and induction of ISG56 expression (Fig. 5c–f). In sharp contrast, treatment of the cells with the two different Nrf2 inducers led to a reduction in ISRE promoter activity, an inca-pacity to phosphorylate STAT1 and to upregulate ISG56 which corroborated with an impaired expression of STING (Fig. 5c–f).

To further examine whether activation of Nrf2 could possibly be a future approach for treating STING-dependent inflamma-tory diseases, we used immortalized fibroblasts from three patients who harbor gain-of-function mutations in STING and who are diagnosed with SAVI[19]. Although harboring a gain-of-function mutation in STING, fibroblasts from SAVI patients do not spontaneously release type I IFN or upregulate ISGs but are highly responsive to STING ligands[19]. Here we observed that treatment with SFN reduced STING expression, STING signaling and release of type I IFN as well as subsequent induction of ISGs in response to stimulation with cGAMP (Fig. 5g–j). Exploiting metabolic reprogramming for treatment of patients with inflammatory diseases, including those that depend on STING, could be an attractive therapeutic strategy. We therefore tested if 4-OI treatment of immortalized fibroblasts from patients diagnosed with SAVI patients was sufficient to affect STING expression and responsiveness to STING agonists in these cells. Here we found that 4-OI inhibited STING expression and subsequently STING signaling in response to cGAMP (Fig. 5k–n).

## Discussion

With this report, we demonstrate that Nrf2 can regulate cytosolic antiviral sensing of DNA and the subsequent release of antiviral type I IFNs by repressing STING expression. This regulation occurred during LPS-induced metabolic reprogramming and hereby suggests that metabolic reprogramming is linked to reg-ulation of antiviral cytosolic DNA sensing. This negative feedback loop might be beneficial to the host since excessive release of type I IFNs is associated with increased pathology in both acute and chronic infection[16,44]. Considering the delay in repression of STING by Nrf2, inhibition of type I IFN by 4-OI might be par-ticularly relevant for reducing prolonged and pathogenic release of type I IFNs in chronic infections. The delay in STING protein degradation was most likely caused by the fact that Nrf2 regulated STING post-transcriptionally. Thus, STING expression levels slowly decreased as STING was gradually degraded through natural cellular protein turn-over processes—while at the same time replenishment by de novo translation was prevented through the effect of Nrf2 on STING mRNA. The inability of ULK1 depletion to rescue Nrf2-mediated STING repression

shows that Nrf2-mediated repression of STING expression is truly distinct from STING-activation-induced degradation, where STING-activation leads to a rapid degradation of STING protein in an ULK1-dependent manner[36]. Autophagy is intricately linked to Nrf2 as Nrf2 regulates the expression of several autophagy proteins including p62/SQSTM1that has been recently involved in the degradation of STING protein[45–47]. However, silencing of the autophagy regulatory proteins ULK1, ATG7, or SQSTM1in A549 and HaCat human cells either had no or a limited effect on Nrf2-mediated STING repression, thus distinguishing Nrf2-mediated repression of STING from autophagy-driven degrada-tion of STING.

Reduction of STING expression by Nrf2 is also mechanistically distinct from how Nrf2 reduces the release of the pro-inflammatory cytokines IL-1β and IL-6. In those cases, Nrf2 bound in the proximity of the respective promoter regions and reduced the recruitment of RNA polymerase II to the genes encoding IL-1β and IL-6[6]. Here, we report that *TMEM173* mRNA is quite stable compared to other immune-related genes like *MB21D1* and weakly rely on RNA Polymerase II regulation in human A549 cells. Interestingly, we demonstrate that Nrf2 regulated STING levels by controlling *TMEM173* stability. The ability of the Nrf2 inhibitor ML385, which works by blocking Nrf2-DNA binding[35], to abrogate Nrf2-mediated suppression of STING suggests that the effect of Nrf2 on STING is not direct but is likely to work through a Nrf2-induced mediator.

Interestingly, although type I IFN responses might be affected by Nrf2 in mice, the effect of Nrf2 on STING expression seemed to be restricted to human cells as no alteration of STING was observed in Nrf2/Keap1 knockout murine cells. Secondary structure of the *TMEM173* gene or STING mRNA, species-specific orthologue proteins involved in the regulation of STING mRNA or predominance of a specific biochemical degradation pathway like autophagy in murine vs human cells, are all possible elements that will have to be taken into consideration in a future exploration of the difference observed in the regulation of STING in human vs murine cells.

Finally, our demonstration that treatment with a cell-permeable derivative of itaconate was sufficient to reduce STING-dependent release of type I IFNs from SAVI-derived fibroblasts promotes the idea that Nrf2 is a valid target in STING-associated inflammatory disorders. Adopting STING-dependent interferonopathies as diseases that could be treated like metabolic diseases could have far reaching consequences and might possibly lead to novel therapeutic strategies for these hard to treat diseases. In conclusion, we have identified a link between metabolic reprogramming and control of cytosolic antiviral sensing. This link depends on the transcription factor Nrf2, which represses STING expression through destabilization of STING mRNA. Our findings add important knowledge to how release of type I IFNs is mechanistically regulated.

## Methods

**Cell lines and culture conditions**. Human lung adenocarcinoma epithelial A549 cells (Sigma-Aldrich), immortalized human HaCat keratinocytes (Thermo Fisher Scientific) and human embryonic kidney HEK293T cells (Sigma-Aldrich) were kindly provided by Søren R. Paludan (Aarhus University, Denmark) and cultured in DMEM (Lonza) supplemented with 10% heat inactivated fetal calf serum, 200 IU mL$^{-1}$ penicillin, 100 µg mL$^{-1}$ streptomycin, and 600 µg mL$^{-1}$ glutamine (hereafter termed DMEM complete). HEK-Blue IFN α/β cells (Invivogen) were cultured in DMEM supplemented with 10% heat inactivated fetal calf serum, 200 IU mL$^{-1}$ penicillin, 100 µg mL$^{-1}$ streptomycin, 600 µg mL$^{-1}$ glutamine, 100 µg mL$^{-1}$ normocin (Invivogen), 30 µg mL$^{-1}$ blasticidin (Invivogen) and 100 µg mL$^{-1}$ zeocin (Invivogen). HEK293 cells stably expressing STING was a kind gift from Long Yang (McGill University). Human acute monocytic leukemia cell line (THP1) was obtained from Martin R. Jakobsen (Aarhus University, Denmark) and cultured in RPMI 1640 (Lonza) supplemented with 10% heat inactivated fetal calf serum, 200 IU mL$^{-1}$ penicillin, 100 µg mL$^{-1}$ streptomycin, and 600 µg mL$^{-1}$

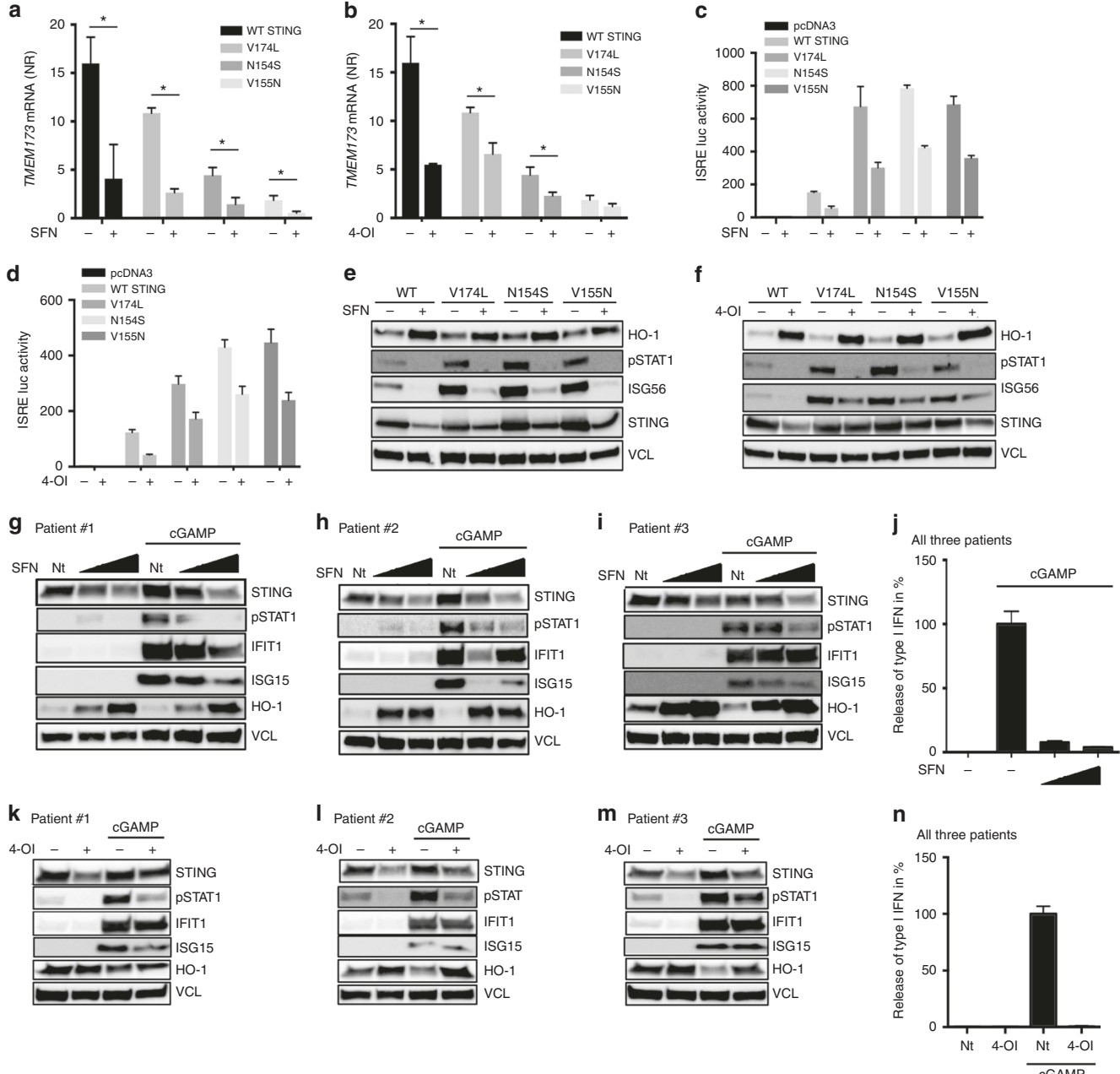

**Fig. 5** Activation of Nrf2 suppresses STING-dependent type IFN response in fibroblasts from SAVI patients. **a**, **b** HEK293T were transiently transfected with human WT and SAVI-STING plasmids (V174L, N152S, or V155M) (500 ng/mL). Cells were treated 1 h prior transfection with SFN (10 μM) or 4-octyl-itaconate (4-OI) (200 μM). *TMEM173* mRNA levels were assessed by qPCR. Data are the means ± s.e.m. of one experiment performed in quadruplicate. **c**, **d** HEK293T were transiently transfected with human WT and SAVI-STING plasmids (V174L, N152S, or V155M) (500 ng/mL), together with an ISRE-Luciferase reporter. Cells were treated 1 h prior transfection of the plasmids with SFN (10 μM) or 4-octyl-itaconate (4-OI) (200 μM). Luciferase activity was assessed 24 h after transfection. Data are the means ± s.e.m. of one experiment performed in duplicate (**c**) and triplicate (**d**). **e**, **f** Whole-cell lysates from (**c**) and (**d**) were analysed by immunoblotting for antiviral signaling with Vinculin (VCL) as loading control. **g–i** Fibroblasts from SAVI patients were pre-treated with increasing doses of L-sulforaphane (SFN) (10 and 20 μM) for 72 h before getting challenged with cGAMP (4 μg/mL). Whole-cell lysates were analysed for late antiviral signaling events by immunoblotting after 24 h of stimulation. Data are from three individual patients. **j** Supernatants from (**g–i**) were collected after 24 h of cGAMP treatment and analysed for type I IFN production. Data are from three individual patients and bars indicated mean and SEM. **k–m** Immortalized fibroblasts from SAVI patients were pre-treated with 4-OI (200 μM) for 48 h before treating with cGAMP. After 24 h of stimulation cells were collected for western blotting (**k–m**), and type I IFN analysis by bioassay (**n**). Data display results from three individual SAVI patients

glutamine (hereafter termed RPMI complete). To differentiate THP1 cells into adherent macrophages, cells were stimulated with 100 nM phorbol 12-myristate 13-acetate (PMA, Sigma-Aldrich) for 24 h in RPMI complete before medium was refreshed and cells allowed to further differentiate for an additional day of culture. The leukemic plasmacytoid dendritic cell line PMDC05 was obtained from Martin R. Jakobsen (Aarhus University, Denmark) and cultured in RPMI (Lonza) supplemented with 10% heat inactivated fetal calf serum, 200 IU mL$^{-1}$ penicillin,

100 μg mL$^{-1}$ streptomycin, and 600 μg mL$^{-1}$ glutamine. Fibroblasts isolated from three different SAVI patients and bearing the same STING mutation were kindly provided by Raphaela Goldbach-Mansky (NIH, Bethesda, USA) and cultured in DMEM (Lonza) supplemented with 10% heat inactivated fetal calf serum, 200 IU mL$^{-1}$ penicillin, 100 μg mL$^{-1}$ streptomycin, and 600 μg mL$^{-1}$ glutamine. All cell lines were regularly tested for mycoplasma contamination by sequencing from GATC Biotech (Germany).

**Primary fibroblast cell lines derived from SAVI patients' superficial skin biopsies.** Patients with genetically confirmed SAVI were enrolled into the protocol (ClinicalTrials.gov NCT02974595) at the National Institutes of Health between 2008 and 2015. The protocol was approved by the National Institute of Allergy and Infectious Diseases (NIAID) IRB at the NIH. Written informed consent was obtained from all participating patients or their legal guardians (RGM). Superficial research biopsies were obtained and primary fibroblast cell lines were generated.

**Primary cells and culture conditions.** Peripheral blood mononuclear cells (PBMCs) were isolated from healthy donors (blood donors gave written consent as accordingly to the ethical guidelines at Aarhus University Hospital) by Ficoll Paque gradient centrifugation (GE Healthcare). Monocytes were separated using a monocyte enrichment kit (STEMCELL) according to the manufacturer's instructions or from PBMCs by adherence to plastic in RPMI 1640 supplemented with 10% AB-positive human serum. Differentiation of monocytes to macrophages was achieved by culturing in Dulbecco's Modified Eagle Medium (DMEM) supplemented with 10% heat inactivated AB-positive human serum 9 days in the presence of 10 ng/ml M-CSF (R&D Systems). Bone marrow cells were obtained from Nrf2 knockout animals purchased from the Jackson laboratory and were subsequently differentiated in macrophages using the following protocol. The cells were differentiated over 6 days using 20–40% L929 supernatant (M-CSF producing murine fibroblast cell line) in RPMI (Lonza) supplemented with 10% heat-inactivated FCS (Sigma-Aldrich), 100 U ml$^{-1}$ penicillin, 100 µg ml$^{-1}$ streptomycin, and 292 µg ml$^{-1}$ L-glutamine (Gibco).

**Drugs, cytokine, and plasmids.** L-sulforaphane (SFN) was purchased from Santacruz Bio and dissolved in DMSO (Sigma-Aldrich). N-acetyl-L-cysteine (L-NAC) was purchased from Santacruz Bio and dissolved in water. pH of the L-NAC solution was adjusted to 7. ML385 was purchased from Axon MedChem and dissolved in DMSO. Interferon-α (A2) was obtained from PBL Assay Science. 4-octyl-itaconate (4-OI) was chemically synthetized by Thomas B. Poulsen (Aarhus University, Denmark) and was dissolved in DMSO (see dedicated section for chemical synthesis). TLR agonists including gardiquimod, LPS, CpG DNA, and Poly (I:C) were all obtained from Invivogen and resuspended in water as per manufacturer's recommendations. Construction of the human Nrf2 overexpression plasmid is described here[48].

**Short-interfering RNA (siRNA)-mediated knockdown.** For short interfering RNA experiments, A549 or HaCat cells were transfected in six-well plates with 80 pmol of human Nrf2(1) (sc-37030), Nrf2(2) (sc-44332), Keap1 (sc-43878), STING (sc-92042), HO-1 (sc-35554), Atg7 (sc-41447), ULK1 (sc-44182), p62/SQSTM1 (sc-29679), IFN α/β R α chain (sc-35637), IFN α/β R β chain (sc-40091) or control si RNA (sc-37007) diluted in serum and antibiotic free DMEM and using Lipofectamine RNAi Max as per manufacturer's instructions. A549 were incubated for 48 h and HaCat cells 72 h in the presence of the siRNA before being processed.

For the Nrf2-gene RNAi screening experiment, A549 cells were transfected in six-well plates with 80 pmol of a cherry-pick RNAi library from Dharmacon. siRNA sequences were diluted in serum and antibiotic free DMEM and transfected using Lipofectamine RNAi Max as per manufacturer's instructions. Cells were incubated for 48 h before being assessed for STING expression by immunoblotting.

For gene interference in primary human monocyte-derived macrophages (hMDMs), cells were transfected on day 6 and 8 post isolation in 48-well plates with Nrf2 specific siRNAs (sc-37030), or siRNA controls (30 nM) using Lipofectamine RNAiMax (Life technologies) according to the manufacturer's instructions, followed by stimulation on day 10.

**Transduction of A549 cells and RAW 264.7 murine macrophages with lentiviral CRISPR/Cas9.** A549 and RAW 264.7 cells deficient in Nrf2/Keap1 were generated using CRISPR/Cas9 system, more specifically by employing the pLentiCRISPR/Cas9 V2 vector (*Addgene*) and guide RNA (gRNA) sequences consisting of gene-specific CRISPR RNA – hNRF2 (T3 5′-GCTGAAAACTTCGAGATATA-3′), hKEAP1 (T1 5′-GGGCGGGCTGTTGTACGCCG-3′, T2 5′-CTGGAGTCGG TGTTGCCGTC-3′), mNrf2 (T1 5′-GACTTGGAGTTGCCACCGCC-3′), mKeap1 (T2 5′-GGCATACATCACTGCGTCCC-3′). For controls, the pLentiCRISPR/Cas9 V2 vector was used without a gRNA sequence. Guides were cloned into the vector and infectious lenti-virions were produced by transfecting HEK293T cells with pLentiCRISPR/Cas9 V2 vector, pMD.2G (*Addgene*), pRSV.Rev (Addgene) and pMDlg/p-RRE (Addgene). Knockout of the targeted gene-products were validated by immunoblotting or other functional assays.

**Genomic deletion of Nrf2 peak region in A549 cells.** To achieve maximum knockout efficiency for the dual cutting for knockout, we designed two-specific gRNAs on each site of region of interest (NRF2-L-T1: 5′-GCCGGTGACTTCA TCCGGCC-3′; NRF2-L-T2: 5′-AGAGCAGCGCCCTCTAGCGG-3′; NRF2-R-T1: 5′-AACCAGACGCCATGCCCGTC-3′; NRF2-R-T2: 5′-CTACCCTCTCTCAGA CCAAC-3′). NRF2-R-T1: 5′-AACCAGACGCCATGCCCGTC-3′; NRF2-R-T2: 5′-CTACCCTCTCTCAGACCAAC-3′). To generate the gRNA expression vectors, complementary gRNA oligos were annealed in 1X NEB buffer 2, cloned into the lentiGuide-Puro plasmid (a gift from Feng Zhang, Addgene plasmid #52963), and

subsequently validated by Sanger sequencing. To generate the gRNA expression vectors, complementary gRNA oligos were annealed in 1X NEB buffer 2, cloned into the lentiCRISPRv2 plasmid (a gift from Feng Zhang, Addgene plasmid #52961), and subsequently validated by Sanger sequencing.

**dsDNA and cGAMP stimulation of cells.** HSV-60 naked, a viral dsDNA motif and 2′3′-cGAMP, a STING ligand, were both obtained from Invivogen. Intracellular delivery of dsDNA and cGAMP was achieved using Lipofectamine 2000 (Invitrogen) diluted in serum-free medium with a ratio of Lipo.dsDNA/cGAMP of 1:1. Final concentration for both dsDNA and cGAMP was 4 µg mL$^{-1}$.

**HSV production, quantification, and infection.** HSV-1 KOS strain expressing GFP (HSV-1–GFP), HSV-2 333 strain and HSV-2 MS strain were kindly provided by Søren Paludan (Aarhus University, Aarhus, Denmark). All HSVs were propagated in Vero cells, purified by ultra-centrifugation, and titrated by plaque assay. A549 and HaCat cells were infected with the different HSVs at a multiplicity of infection (MOI) of 0.01 in a small volume of serum-free medium for 1 h at 37 °C. Prior to analysis, cells were incubated with DMEM complete for an additional day of culture.

**Western blot.** Four hundred thousand THP1 cells and 1 million A549 or HaCat cells were lysed in 100 µL or 60 µL of ice-cold Pierce RIPA lysis buffer (Thermo Scientific) supplemented with 10 mM NaF, 1X complete protease cocktail inhibitor (Roche) and 5 IU mL$^{-1}$ benzonase (Sigma), respectively. Protein concentration was determined using a BCA protein assay kit (Thermo Scientific). Whole-cell lysates were denatured for 3 min at 95 °C in presence of 1X XT Sample Buffer (BioRad) and 1X XT reducing agent (BioRad). A total of 10–40 µg of reduced samples was separated by SDS-PAGE on 4–20% Criterion TGX precast gradient gels (BioRad). Each gel was run initially for 15 min at 70 V and 45 min at 120 V. Transfer onto PVDF membranes (BioRad) was done using a Trans-Blot Turbo Transfer system for 7 min. Membranes were blocked for 1 h with 5% skim-milk (Sigma-Aldrich) at room temperature in PBS supplemented with 0.05% Tween-20 (PBST). Membranes were fractionated in smaller pieces and probed overnight at 4 °C with any of the following specific primary antibodies in PBST: anti-Nrf2 (12721, Cell Signaling 1:1000), anti-MDA5 (5321, Cell Signaling 1:1000), anti-MAVS (3993, Cell Signaling 1:1000), anti-TBK1/NAK (3013, Cell Signaling 1:1000), anti-phospho-TBK1/NAK (5483, Cell Signaling 1:1000), anti-MyD88 (4283, Cell Signaling 1:1000), anti-NF-kB p65 (8242, Cell Signaling 1:1000), anti-Atg7 (8558, Cell Signaling 1:1000), anti-ULK1 (8054, Cell Signaling 1:1000), anti-SQSTM1/p62 (8025, Cell Signaling 1:1000), anti-RIG-I (3743, Cell Signaling 1:1000), anti-IRF3 (11904, Cell Signaling 1:1000), anti-phospho-IRF3 (4947, Cell Signaling 1:500), anti-STAT1 (9172, Cell Signaling 1:1000), anti-phospho-STAT1 (7649, Cell Signaling 1:1000), anti-HO-1 (5853, Cell Signaling 1:1000), anti-IFIT1 (14769, Cell Signaling 1:1000), anti-Nrf2 (12721, Cell Signaling 1:1000), anti-cGAS (15102, Cell Signaling 1:1000), anti-ISG15 (2758, Cell Signaling 1:1000), anti-STING (13647, Cell Signaling 1:1000), anti-IFI16 (sc-8023, Santacruz Bio 1:1000), anti-NqO1 (3187, Cell Signaling 1:1000), anti-MDM2 (86934, Cell Signaling 1:1000), anti-Cyclin B1 (12231, Cell Signaling 1:1000), anti-HSV-1–2 VP5 (ab6508, AbCam 1:1000) and anti-Vinculin (18799, Cell Signaling 1:1000) used as loading control. After three washes in PBST, secondary antibodies, peroxidase-conjugated F(ab)2 donkey anti-mouse IgG (H+L) (1:10000) or peroxidase-conjugated F(ab)2 donkey anti-rabbit IgG (H+L) (1:10000) (Jackson Immuno Research) were added to the membrane in PBST 1% milk for 1 h at room temperature. All membranes were washed three times and exposed using either the SuperSignal West Pico PLUS chemiluminescent substrate or the SuperSignal West Femto maximum sensitivity substrate (Thermo Scientific) and an Image Quant LAS4000 mini imager (GE Healthcare). The levels of proteins were quantified by densitometry using the Image J software.

Uncropped images of the western blots are provided in Supplementary figure 15.

**Semi-native WB dimerization assay.** IRF3, phosphor-IRF3 and STING dimerization was assayed under semi-native conditions. Cells were lysed in ice-cold Pierce RIPA lysis buffer (Thermo Scientific) supplemented with 10 mM NaF, 1X complete protease cocktail inhibitor (Roche) and 5 IU mL$^{-1}$ benzonaze (Sigma). Protein concentration was determined using a BCA protein assay kit (Thermo Scientific). Whole-cell lysates were mixed with 1X XT Sample Buffer (BioRad); samples were neither reduced nor heated before separation was done on 4–20% Criterion TGX precast gradient gels (BioRad) by SDS-PAGE electrophoresis. Each gel was run initially for 15 min at 70 V and 15 min at 120 V. Transfer onto PVDF membranes (BioRad) was done using a Trans-Blot Turbo Transfer system for 7 min. Membranes were blocked for 1 h with 5% skim-milk (Sigma-Aldrich) at room temperature in PBS supplemented with 0.05% Tween-20 (PBST). Membranes were probed overnight at 4 °C with any of the following specific primary antibodies in PBST: anti-IRF3 (11904, Cell Signaling 1:1000), anti-phospho-IRF3 (4947, Cell Signaling 1:500), and anti-STING (13647, Cell Signaling 1:1000). After three washes in PBST, secondary antibodies, peroxidase-conjugated F(ab)2 donkey anti-rabbit IgG (H+L) (1:10,000) (Jackson Immuno Research) were added to the membrane in PBST 1% milk for 1 h at room temperature. All membranes were

washed three times and exposed using either the SuperSignal West Pico PLUS chemiluminescent substrate or the SuperSignal West Femto maximum sensitivity substrate (Thermo Scientific).

**qPCR analysis.** Gene expression was determined by real-time quantitative PCR, using TaqMan detection systems (Applied Biosciences). RNA was extracted using the High Pure RNA Isolation kit (Roche) and RNA quality was assessed by Nanodrop spectrometry (Thermo Fisher). RNA levels for human *Ifnb1* (Hs01077958_s1), *Cxcl10* (Hs00171051_m1), *Tmem173* (Hs00736955_g1), *Hmox-1* (Hs01110250_m1), *Nfe2l2 (Nrf2)* (Hs00975961_g1), *Mb21d1 (cGas)* (Hs00403553_m1) and *β-Actin* (Hs01060665_g1) were analysed using premade TaqMan assays and the RNA-to-Ct-1-Step kit according to the manufacturer's recommendations (Applied Biosciences).

**RNAseq and ChIP-seq analysis.** One million of A549 cells silenced or not for Nrf2 using siRNA were snap-frozen on dry ice. RNA extraction, library preparation, RNAseq and bioinformatics analysis was performed at Active Motif (Carlsbad, California, USA). Briefly, the sequencing reads were aligned to human genome (version Hg19)[49] and the resulting binary alignment (BAM) files were used to calculate the gene counts that represent total number of sequencing reads aligned to a gene. To identify differentially expressed genes between control and Nrf2 siRNA-treated samples, DESeq2 algorithm was used[50]. The list of differentially expressed genes from DESeq2 output were selected based on 10% adjusted P-value level and a FDR of 0.1. Among these, the highly significant genes (FDR <10%) genes were selected that are involved in pattern recognition signaling, antiviral signaling and experimentally identified Nrf2 transcriptional targets[51–53]. To depict these genes as heatmap, count data were transformed using regularized-logarithm transformation (rld)[50] and the resulting values were mean-centered and plotted usinged pheatmap package available in bioconductor repository (Raivo Kolde (2015). pheatmap: Pretty Heatmaps. R package version1.0.8. http://CRAN.R-project.org/package=pheatmap). Gene ontology and KEGG pathway enrichment analysis was done using DAVID bioinformatics resources portal[54].

**Chromatin immunoprecipitation (ChIP).** ChIP was performed by Active Motif (Carlsbad, California, USA). A549 cells were fixed with 1% formaldehyde for 15 min and quenched with 0.125 M glycine. Chromatin was isolated by the addition of lysis buffer, followed by disruption with a Dounce homogenizer. Lysates were sonicated and the DNA sheared to an average length of 300–500 bp. Genomic DNA (Input) was prepared by treating aliquots of chromatin with RNase, proteinase K and heat for de-crosslinking, followed by ethanol precipitation. Pellets were resuspended and the resulting DNA was quantified on a NanoDrop spectrophotometer. Extrapolation to the original chromatin volume allowed quantitation of the total chromatin yield.

An aliquot of chromatin (30 µg) was precleared with protein A (for Nrf2) or G (for Pol2) agarose beads (Invitrogen). Genomic DNA regions of interest were isolated using 4 µg of target specific antibody. Complexes were washed, eluted from the beads with SDS buffer, and subjected to RNase and proteinase K treatment. Crosslinks were reversed by incubation overnight at 65 °C, and ChIP DNA was purified by phenol-chloroform extraction and ethanol precipitation.

Quantitative PCR (QPCR) reactions were carried out in triplicate on specific genomic regions using SYBR Green Supermix (BioRad). The resulting signals were normalized for primer efficiency by carrying out QPCR for each primer pair using Input DNA.

**ChIP sequencing (Illumina).** ChIP sequencing was performed by Active Motif (Carlsbad, California, USA). Illumina sequencing libraries were prepared from the ChIP and Input DNAs by the standard consecutive enzymatic steps of end-polishing, dA-addition, and adaptor ligation. After a final PCR amplification step, the resulting DNA libraries were quantified and sequenced on Illumina's NextSeq 500 (75 nt reads, single end). Reads were aligned to the human genome (hg19) using the BWA algorithm (default settings). Duplicate reads were removed and only uniquely mapped reads (mapping quality >=25) were used for further analysis. Alignments were extended in silico at their 3′-ends to a length of 200 bp, which is the average genomic fragment length in the size-selected library, and assigned to 32-nt bins along the genome. The resulting histograms (genomic "signal maps") were stored in bigWig files. For Nrf2, peak locations were determined using the MACS algorithm (v2.1.0) with a cutoff of $p$-value = 1e−7. RNA Pol2-enriched regions were identified using the SICER algorithm at a cutoff of FDR 1E-10 and a max gap parameter of 600 bp. Peaks that were on the ENCODE blacklist of known false ChIP-Seq peaks were removed. Signal maps and peak locations were used as input data to Active Motifs proprietary analysis program, which creates Excel tables containing detailed information on sample comparison, peak metrics, peak locations and gene annotations.

**Measurement of itaconic acid.** All the solvents used for extraction and mobile phase were LC–MS Optima Grade (Fisher Chemical Ltd., Pittsburgh, PA, USA). The quantification of itaconic acid in cell cultures was determined using standard dilutions of itaconic acid. The stock solution of itaconic acid (Sigma-Aldrich I29204, St. Louis, MO, USA) was prepared in water with 0.1% formic acid at the

concentration of 1000 µmol/mL. The calibration curve was prepared by serial diluting 50 µmol/mL to 3.125 µmol/mL in water with 0.1% formic acid resulting in a linear equation using for quantification. Cell culture samples (200 µL supernatant or 100 µL of pellet) were diluted in 800 µL of methanol. After being vortexed, the diluted samples were centrifuged at 16,000×$g$ for 5 min at 4 °C and supernatant were transferred to a new vial and evaporated in a vacuum concentration system (Savant Instruments, Holbrook, NY, USA) and re-dissolved in 100 µL of water 0.1% formic acid. Itaconic acid was analyzed using an Agilent-1290 ultra-performance liquid-chromatography system equipped with autosampler and coupled with an Agilent-6495 triple-quadrupole mass spectrometry (Agilent Technologies, Palo Alto, CA, USA). The injection volume of extraction was 5 µL and the autosampler temperature was 4 °C. The separation was performed using an Zorbax Eclipse column C18 (2.1 × 50 mm × 1.8 µm; Agilent Technologies) at 0.5 mL min and 40 °C. The mobile phase A was 0.1% formic acid in water and mobile phase B 0.1% formic acid in acetonitrile. The gradient of mobile phase A starting at 75%, maintained for 0.2 min, followed by 50% at min 2 to reach the initial state at min 2.2 followed by 0.8 min of re-equilibration. The mass spectrometry was performed in negative ionization mode. The capillary voltage and nebulizer were set at 1500 V and 11 V, respectively. The electro-spray ionization source was operated at 290 °C with ultrahigh-purity nitrogen as cone gas (11 L/min). The desolvation temperature was 400 °C and the gas flow set at 12 L/min. Nitrogen was used as collosion gas, and the collosion energy was set at 5 eV for itaconic acid. Multiple reactions monitoring (MRM) was applied to detect the analyte by the following ion transitions (m/z): Itaconic acid (129.09), 89.1.

The data acquisition, process, and quantification were performed using MassHunter software (Agilent Technologies).

**ROS production.** Total ROS accumulation was evaluated by flow cytometry using the H2DCFDA redox sensitive probe. Following siRNA treatment and/or treatment with the appropriate controls, cells were washed in PBS before incubation with the probe (1 µM) for 30 min at 37 °C. After incubation, cells were washed twice in PBS before FACS analysis using a Novo Cyte flow cytometer. FlowJo software was used to analyze the data.

**Type I IFN assay.** Functional type I IFN was quantified using the reporter cell line HEK-Blue IFN-α/β according to the manufacturer's instructions (Invivogen).

**Luciferase assay.** For ARE-Luciferase assays, experiments were performed using the calcium phosphate transfection method. For Are luciferase assays, HEK293T cells were transfected with 50 ng of pRLTK reporter plasmid, 100 ng of pP-ARE-Luc reporter, and increasing doses of the expression plasmid encoding for Nrf2 or STING, along with the appropriate amount of empty vector. After 24 h of transfection and stimulation, luciferase activity was measured with a dual-luciferase reporter assay and a GloMax 20/20 luminometer.

**Co-immunoprecipitation.** HEK293T cells were collected after transfection with the different overexpression plasmids and resuspended in 300 µl Pierce IP lysis buffer (Thermo Scientific) supplemented with 1X Complete Ultra (Roche) and NaF 0.5 mM. Cells were allowed to lyse at 4 °C for 90 min under rotation and cytosolic supernatants were cleared by centrifugation at 3000×$g$ for 10 min. A volume of 200 µl of cleared lysates were then incubated at 4 °C overnight with an antibody against STING (Cell Signaling 1:50). On the following day, each lysate was incubated with pre-washed Dynabeads magnetic protein G (Invitrogen) for 2 h at 4 °C. Immunoprecipitated complexes were eluted from the beads with a glycine buffer (200 mM glycine, pH 2.5), and protein expression was evaluated by western blotting.

**Polysome profiling and quantitative PCR.** Cells seeded in 10 cm plates cells were the following day transfected with control siRNAs or siRNAs targeting KEAP, using Lipofectamine RNAi (Life Technologies) transfection agent according to manufacturer's directions. A total of 46 h after transfection media were replenished with fresh DMEM, 10% FBS and cells cultured for an additional 2 h. Cells were then treated with 100 µg/mL cycloheximide for 3 min followed by two washes in ice-cold PBS containing 100 µg/mL cycloheximide. Cells were subsequently lysed directly in the cell culture plate in 1000 µL of 1X polysome extraction buffer (1× PEB; 50 mM Tris-HCl at pH 7.4, 150 mM NaCl, 7.5 mM MgCl$_2$, 0.5% Triton X-100, 500 µg/mL Heparin, 100 µg/mL cycloheximide), scraped off, and incubated for 5 min on ice. Lysates were cleared by centrifugation at 15,300×$g$, 4 °C for 10 min. Complexes were separated in 10 mL of linear 1X PEB/10–50% sucrose gradients by ultra-centrifugation at 150,000×$g$ (35,000 rpm) in a Beckman SW41 rotor for 2.5 h at 4 °C. Twelve 1-mL fractions were collected using a Teledyne/ISCO fractionation system. Five-hundred microliters from each fraction was vortexed for 20 s after addition of 750 µL of 8M guanidine-HCl and 750 µL of isopropanol and precipitated overnight at −20 °C. Precipitates were isolated by centrifugation at 17,900×$g$ for 30 min at 4 °C, and pellets were subsequently dissolved in Trizol solution (Invitrogen). RNA was isolated according to the manufacturer's protocol except for one additional 300 µl chloroform extraction prior to isopropanol precipitation. Pellets were dissolved in 20 µl nuclease-free water (Ambion) and equal volumes from the fractions were used for quantitative PCR. First-strand cDNA

synthesis was carried out using equal volumes of RNA and the Maxima First Strand cDNA Synthesis Kit for qPCR (Thermo Fisher Scientific) according to manufacturer's protocol. STING mRNA expression levels were measured using the Maxima Probe qPCR Master Mix (Thermo Fisher Scientific) and TaqMan primers and probes (Hs00736955_g1, Thermo Fisher Scientific) according to manufacturer's protocol. GAPDH mRNA expression levels were measured using Platinum SYBR Green qPCR Supermix-UDG (Invitrogen) according to manufacturer's protocol and gene-specific primers (GAPDH (s): 5′-GTCAGCCGCATCTTCTT TTG-3′, GAPDH (as): 5′-GCGCCCAATACGACCAAATC-3′). An AriaMx Real-time PCR System (Agilent Technologies, CA, California, United States) were used for quantification of mRNA levels and the $X_0$ method was used for calculations of relative mRNA levels[55].

**Synthesis of 4-octyl-itaconate (4-OI)**. See supplementary methods.

**Statistical analysis**. Values were expressed as the mean ± SEM. Graphs and statistics were computed using Graph Pad Prism 7. An unpaired, two-tailed Student's t-test was used to determine significance of the difference between the control and each experimental condition. P-values of <0.05 were considered statistically significant, ***$p < 0.001$; **$p < 0.01$, and *$p < 0.05$. GSE113522 is the reference Series for both ChIP-seq and RNA seq: [https://www.ncbi.nlm.nih.gov/geo/query/acc.cgi?acc = GSE113522] SubSeries that are linked to GSE113522: [https://www.ncbi.nlm.nih.gov/geo/query/acc.cgi?acc = GSE113497] [https://www.ncbi.nlm.nih.gov/geo/query/acc.cgi?acc = GSE113519]

## Data availability
Sequencing data are uploaded to GEO.

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

## Acknowledgements

This research work was supported by Hørslevsfonden, Agnes and Poul Friis Fond, Brdr. Hartmanns fond, Oda og Hans Svenningsens Fond, Augustinus Fonden and Hede Nielsens Fond, to C.K.H. and by Riisfortfonden, Agnes and Poul Friis Fond, Dagmar Marshalls Fond, Magda Sofie og Aase Lütz's mindelegat Fond, Tømrermester Jørgen Holm og hustru Elisa F. Hansens indelegat Fond, Fabrikant Einar Willumsens mindelegat Fond, Læge Sofus Carl Emil Friis og Hustru Olga Doris Friis Legat to D.O. T.B.P. was supported by a Sapere Aude II Grant from the Independent Research Foundation Denmark (6110-00600B). Salaries were supported by a Carlsbergfonden International Research Fellowship to D.O., a Lundbeck postdoctoral fellowship to M.B.I., a Lundbeck PhD fellowship to C.G., a research year grant from the Danish Research Council to Victor Bruun, and a PhD fellowship to Jacob Thyrsted from the Department of Health Sciences at Aarhus university.

## Author contributions

D.O. and C.K.H. conceived the project, designed the experiments, and wrote the manuscript. D.O. performed most of the experiments and assembled the figures. A.M.B., C.G., R.L., V.B., L.B., D.G.R., B.M., A.K.H., F.P., and J.T. performed in vitro experiments. A.L.T. helped with the A549 CRISPR constructions. C.K. and A.L. performed KO in primary hMDMs. A.L.H. gave her support on flow cytometry experiments. M.B.I. helped with BMM differentiations and mouse work. R.G.-M. provided the fibroblasts from the SAVI patients. K.A.F. and M.M. provided the SAVI plasmids. N.L.V. and T.B.P. generated 4-OI. Y.L. and M.N., S.P., and S.B. performed bioinformatics. H.C.B. and V.M.A. measured itaconate. C.S. performed luciferase experiments. L.Y. generated HEK293 cells stably expressing STING. L.L. generated CrisprCas9 editing of A549 cells. C.K.D and A.K.H. performed polysome analysis. M.R.J. and L.A.O. edited the manuscript. C.K.H. managed the project.

## Additional information

**Competing interests:** The authors declare no competing interests.

