## [Peer Review File · Nature Communications]

Reviewers' comments:

Reviewer #1 (Remarks to the Author):

The manuscript by Oagnier et al reports on the role of Nrf2 in regulation STING pathway activation. It is potentially very interesting observation, however, the work in its current shape lacks the coherency and does not prove any specific point but rather jumps from one potentially interesting observation to the other without making connections or providing explanations.

Here are some major concerns with the work and the manuscript:

1. Introduction:

First, regarding the overview of the previous work described in the introduction - some very critical references and statements are missing: anti-inflammatory action of endogenous itaconate was first proved in work Lampropoulou et al (Cell Metabolism, 2016) by using macrophages from Irg1-deficient mice that lack itaconate production. Furthermore, in the same paper direct biochemical inhibition of Sdh by itaconate was shown and concurrently Cordes et al published the same biochemical observation at the same time. Somehow authors cite Cordes et al, and seem to be unaware of the work of Lampropoulou et al. The authors should read this paper, which is the key work in the field of itaconate immunology and cite it appropriately in the text.

2. Fig. 1:

Data on RNA-seq and CHIP-seq need to be explicitly provided, both as repository deposition (ArrayExpress or GEO) as well as the supplementary tables -

a. tables of differentially expressed genes for the RNA-seq analysis - DESeq2 pipeline that authors used provides very nice table format output that can go directly into supp material, with p-values and log-fold changes.

(also, a note on the accuracy of the manuscript - please fix the citations in the methods section - instead of actual references you have PMIDs written there, which does not reflect well to the accuracy of the work.)

b. tables of the peak calling with the peaks positions and their statistical significance - both MACS2 and SICER software provide nice outputs that can be plugged in directly as supplementary table.

3. Fig.1&2:

The need to put in results of primary analysis is not just a matter of accounting or following the rules. Based on the figure 1b, the fold change for TMEM173 is barely 2-fold, based on the color scheme it looks more like 1.5 fold change (even considering the logarithmic nature of the color scheme). Natural question arises - how many total differentially expressed genes are there upregulated upon siNrf2 and whether TMEM173 is even anywhere on top of the list, or somewhere in the middle.

This is important because protein level difference of STING upon Nrf2 knockdown shown on Fig.1g are significantly higher than 2-fold, suggesting that it is not stability of mRNA but rather translational regulation that is responsible for the build up of STING protein upon Nrf2 knock-down.

To clarify, i do not dispute results on the mRNA stability shown on Fig.2 but if mRNA changes are only 2-fold, and protein changes are 20(?)-, 50(?)- fold, then natural expectation is that major mechanism of regulation of STING protein levels is not through the mRNA stability differences but through some kind of translational regulation.

Authors should consider experiments with proteosomal inhibitors and simple genetic constructs for TMEM173 to be able to claim the exact mechanism of Nrf2-mediated regulation. Given that authors work with the cell line, introducing TMEM173 genetic constructs should be very

straightforward.

If the central statement of work is Nrf2-STING connection, it has not been sufficiently proven, explained or even explored in the current manuscript.

4. Fig.4&5:

To make claims about itaconate, authors use an itaconate derivative - octyl-itaconate - and refer to the paper by O'Neill group that just appeared AOP in Nature last week.

It is a very nice work that dissects effects of octyl-itaconate on immune activation in great details. As far as i can see the paper from O'Neill group was submitted in summer last year while Olganier et al submitted their manuscript just now. During the time between summer 2017 and now, a couple of other major works in the field of itaconate biology have been published. Authors need to familiarize themselves with the field developments that happened, specifically, two of the following works:

First, in the fall 2017, Vamsi Mootha's lab published the work (Shen et al, Cell 2017) that clearly shows that itaconate (in its natural form) penetrates into the cells directly, i.e. it is itself membrane-permeable metabolite.

Second, recent work from Stallings and Diamond labs (Nair et al, J Exp Med, 2018) shows that itaconic acid directly (rather than any itaconate derivative) rescues transcriptional response of Irg1ko macrophages to in vitro MTb infection - also indicating that itaconate is a membrane permeable metabolite in its natural form.

So, as of March 2018, when talking about natural itaconate (TCA-cycle derived metabolite, as authors put it), there is no need to employ various chemical derivatives - itaconate itself can and SHOULD be used directly. Given the results from Mootha and Diamond labs, it is scientifically inappropriate to talk about itaconate in the manuscript and figure caption (even showing itaconate on the schematic panel Fig.4g) while presenting experimental data only on octyl-itaconate treatment. Hence, either authors have to reformulate the paper to explicitly state that octyl-itaconate inhibits STING pathway, or to show results with the actual physiological itaconate or Irg1ko to support the claims made in current write-up.

Reviewer #2 (Remarks to the Author):

The authors first observe that targeting of Nrf2 in human cell lines impacts STING expression. They next observe that Nrf2 impacts TMEM173 mRNA stability over time and ultimately impacts innate viral immunity and virus production. The response is subsequently linked to the induction of Nrf2 by itaconate esters. Finally, using STING mutant SAV1 cells the authors also demonstrate the impact of Nrf2 activation in this system.

This is an interesting paper that presents a distinct regulatory mechanism of Nrf2 on STING levels. The novelty lies in a new mechanism for Nrf2 regulation of STING/innate immunity. However, given the pleiotropic effects of Nrf2 on cell pathways and the specific conditions under which they observe their effect, some additional studies are warranted. For example, the authors demonstrate that TMEM173 expression is destabilized by Nrf2 beyond 4 hours, but no reason for this kinetic change was presented as I read. Is the effect direct or indirect through expression of another protein? Therefore, the phenotypes and potential impact are strong but a definitive mechanism is not clear.

1. Targeting of Nrf2 in cancer cell lines significantly alters their biosynthesis and growth capacity, and this presumably translates to compromised viral growth as well. The claim that Nrf2 impairs innate immunity is therefore somewhat strong. Some additional data should be provided on cell

growth to highlight this discrepancy and the language toned down. Alternatively, including controls that show how targeting STING levels independently impacts viral titer (or does not) would be helpful.

2. The authors routinely use the term itaconate when referring to the OI ester (particularly in subtitles). These molecules are different (and behave differently in cells as demonstrated by the Mills paper), so when using OI it should be referred to as such. The same applies to the drug DFO and its unesterified counterpart fumarate. Revision is necessary throughout the manuscript.

3. Along these lines, it would be informative to investigate how Irg1 KD or KO impacts TMEM173 mRNA stability over time.

4. In Figure 4b the authors show that unstimulated PMDC05 cells have mM levels of itaconate in supernatant (unclear what "supernatant" of cells even is – medium?). This indicates that TLR4 signaling is strongly active even basally. Please clarify as most unstimulated cells do not produce so much itaconate and define supernatant. Furthermore, the fold change in levels should be orders of magnitude rather than 2 fold.

Reviewer #3 (Remarks to the Author):

In this interesting well-designed and executed work by OLAGNIER et al. the authors present a novel role for the transcription factor Nrf2: the modulation of cytosolic DNA sensing and the concomitant inflammatory response by regulating the STING expression. This role is exerted upon induction of immune/inflammatory response that leads to increased itaconate levels that was recently shown (appended publication) to activate Nrf2 pathway. These results, even though limited so far to cell cultures (primary and cell lines), would certainly be of interest to scientists working in the field of immunity and metabolism and could potentially pave the way for further basic research in the area and for clinical trials using Nrf2 pathway activators in patients with STING-dependent interferonopathy and relevant disorders. One limitation is that the mechanism through which Nrf2 affects mRNA stability of STING has not been addressed and is left for future studies. The following points need to be considered by the authors in the order of appearance in the paper.

1. Fig. 1c,d,e,f: What do the individual points show? Independent samples of an experiment or independent experiments? This is not totally clear from the figure legend.

2. Fig. 1m,4q and S7. The protein levels of the partly Nrf2 regulated gene HO-1 (PMID: 17942419) from western blots are shown at the representative photos to be increased after siNrf2 treatment. It is understandable that its baseline levels are very low and this could be due to variability but please explain this based on all your results. It would be advisable to use other Nrf2 target genes as well for validation purposes such as Nqo1, Txnrd1 etc.

3. Lines 180-182. The results from ChIP-Seq are very interesting and combined with the mRNA levels of STING point to the direction that indeed Nrf2 should affect mRNA stability. Authors should consider changing the expression from "...a novel post-translational..." to "... a novel post-transcriptional...".

4. Lines 205-206. In general, it is a bit confusing jumping from one cell line to another during the experimental results. The authors should clearly state the advantages of using a specific cell line for a certain experiment like they did for example with A549. In this case it is not so evident why they picked the PMDC05 line.

5. Fig. 4h. It is not acceptable to present in a final paper these results without enough n number to perform statistics.

6. The point that Nrf2 can be the mediator of STING repression by itaconate is novel and very interesting to the field and is proven by siRNA studies (mainly shown in Fig.4). As these results are central to this work, it would be advisable to use at least one more alternative method of silencing Nrf2 (e.g. Crispr as in the other experiments or another siRNA construct) to ensure that these results can be replicated and reduce the possibility of seeing off target effects. Moreover, in line 268 it would be better to use the expression "is mostly regulated by the transcription factor Nrf2..." as silencing Nrf2 does not always lead to complete reversal of the endpoints such as TMEM173 expression (Fig. 4o,p).

Point-by-point reply

Reviewer #1

1. Introduction:

First, regarding the overview of the previous work described in the introduction - some very critical references and statements are missing: anti-inflammatory action of endogenous itaconate was first proved in work Lampropoulou et al (Cell Metabolism, 2016) by using macrophages from Irg1-deficient mice that lack itaconate production. Furthermore, in the same paper direct biochemical inhibition of Sdh by itaconate was shown and concurrently Cordes et al published the same biochemical observation at the same time. Somehow authors cite Cordes et al, and seem to be unaware of the work of Lampropoulou et al. The authors should read this paper, which is the key work in the field of itaconate immunology and cite it appropriately in the text.

This is certainly a valid point. We have altered the text, so that it emphasizes, and cites, the important work by Lampropoulou, as suggested by the reviewer.

2. Fig. 1:

Data on RNA-seq and CHIP-seq need to be explicitly provided, both as repository deposition (ArrayExpress or GEO) as well as the supplementary tables –

Links to the RNA-seq and ChIP-seq data deposited in GEO is now included in the methods and materials section.

Tables of differentially expressed genes for the RNA-seq analysis - DESeq2 pipeline that authors used provides very nice table format output that can go directly into supp material, with p-values and log-fold changes.

To address this point, we have added a supplementary table, containing the DESeq2 values for the RNAseq analysis (Table 1)

(also, a note on the accuracy of the manuscript - please fix the citations in the methods section - instead of actual references you have PMIDs written there, which does not reflect well to the accuracy of the work.).

The PMIDS were somehow not converted to regular citations in the original manuscript. This has now been changed to regular citations.

Tables of the peak calling with the peaks positions and their statistical significance - both MACS2 and SICER software provide nice outputs that can be plugged in directly as supplementary table.

As suggested by the reviewer we have added a supplementary table, which holds a table of the Peak calling and the accompanying meta-data for both the Nrf2 and the Pol II ChIP seq (Table 2 and -3).

3. Fig.1&2:

The need to put in results of primary analysis is not just a matter of accounting or following the rules. Based on the figure 1b, the fold change for TMEM173 is barely 2-fold, based on the color scheme it looks more like 1.5 fold change (even considering the logarithmic nature of the color scheme). Natural question arises - how many total differentially expressed genes are there upregulated upon siNrf2 and whether TMEM173 is even anywhere on top of the list, or somewhere in the middle.

To address this question, we have provided a new supplementary figure displaying the number of differentially regulated genes and a new heat-map (Fig. S1).

We also thank the reviewer for pointing out the issue in interpreting the fold changes in figure 1b heatmap. We apologize for this error in labeling the heatmap legend as log-fold induction. It should be noted that the color scale represents expression values of genes on log2 scale but not fold change. Since fold change for a gene cannot be represented as heatmap across replicates, we only depicted the expression values on log2 scale. To better display the importance of Nrf2 on STING mRNA levels we have generated graphs displaying the effect of Nrf2 silencing on the RNAseq-generated TMEM173 and Nrf2 RNA reads (Fig. 1c).

We also edited our manuscript to show that '1511 genes are differentially expressed (p-value < 0.001 and Fold change cut off > 1 or <-1) between NRF2-KO and WT cells. We have included a waterfall plot to show the position of TMEM173 on fold change rank order. In the figure, red line show TMEM173 (Fig. S1).

This is important because protein level difference of STING upon Nrf2 knockdown shown on Fig.1g are significantly higher than 2-fold, suggesting that it is not stability of mRNA but rather translational regulation that is responsible for the buildup of STING protein upon Nrf2 knock-down. To clarify, I do not dispute results on the mRNA stability shown on Fig.2 but if mRNA changes are only 2-fold, and protein changes are 20(?)-, 50(?) fold, then natural expectation is that major mechanism of regulation of STING protein levels is not through the mRNA stability differences but through some kind of translational regulation.

We agree with the reviewer that the fold increase in STING protein is greater than it is in STING RNA – although not 20-50-fold as suggested by the reviewer. We have used image-J to quantify STING staining in multiple western blots and come up with a difference in STING protein staining (measured colorimetric increase) at approximately 4-5-fold. This quantification is now displayed as Fig1k. This relationship between mRNA changes and changes in protein does not seem implausible to us.

Nevertheless, we agree with the reviewer that the effect Nrf2 has on STING expression might not be entirely due to regulation of mRNA stability. As mRNA stability and efficiency of translation is often interconnected we investigated whether the Nrf2 pathway affects the association of STING mRNA with polysomes. These data did not indicate that translation efficiency of STING mRNA into protein is significantly affected by silencing of Keap1. These data are now displayed in Fig. S11. So, although there might still be unknown factors/pathways that could contribute to the Nrf2 mediated repression of STING we believe that we have now generated sufficient data to suggest that regulation of mRNA is a significant contributor to this phenomenon.

Authors should consider experiments with proteasomal inhibitors and simple genetic constructs for TMEM173 to be able to claim the exact mechanism of Nrf2-mediated regulation. Given that authors work with the cell line, introducing TMEM173 genetic constructs should be very straightforward.

This is a very valid comment and we have therefore conducted additional experiments to address the point. We have used a proteasomal inhibitor to test if proteasomal degradation activity is a strong predictor of STING expression. Here we compared with levels of the proteins cyclin B1 and MDM2, which are both sensitive to proteasomal inhibition. Although there was a slight increase in STING expression, this increase was far below the one observed in the experiments where Nrf2 was silenced and to what was observed with cyclin B1 and MDM2. These data are presented in the new Fig. S9.

Further, as suggested by the author we have used a HEK293-based experiment to demonstrate that also in this system, the overexpression of Nrf2 leads to repression of STING expression. As STING expression is under the control of vector-encoded promoter and not its own, these data further support that the regulation of STING by Nrf2 occurs post-transcriptionally. These data have been added to the manuscript in Fig. S5.

If the central statement of work is Nrf2-STING connection, it has not been sufficiently proven, explained or even explored in the current manuscript.

We hope that with the addition of more experiments we have now satisfied this point. In summary, we have demonstrated that Nrf2 represses STING expression in three unrelated cell lines (one cancer derived, one keratinocyte-like, and one monocyte derived), in primary human MDMs, and in a HEK293-cell based overexpression system. Further, experiments with known activators of Nrf2 and experiments where we silenced the Nrf2 repressor Keap1 also support this finding. Further, we performed a new analysis of our original ChIPseq data using a looser p-value restriction (p-value $1e-3$) to determine if the former and more strict analysis (p-value $1e-5$) had wrongfully disregarded an important Nrf2 peak. Here we noticed a small Nrf2 peak positioned at a distance of approximately 25-30 kilo-bases from *TMEM173* gene. This peak was not called in the original analysis where we used the p-value cutoff of $1e-5$. As this peak might possibly be important for Nrf2 regulation of STING, we created three A549 cell clones with CrisprCas9 generated excisions of this region. For all clones the excision led to a decrease rather than an increase in STING expression, thus supporting that the effect of Nrf2 on STING expression is not mediated through direct regulation of transcription. These data are displayed in Fig S10.

4. Fig.4&5:

To make claims about itaconate, authors use an itaconate derivative - octyl-itaconate - and refer to the paper by O'Neill group that just appeared AOP in Nature last week.

It is a very nice work that dissects effects of octyl-itaconate on immune activation in great details. As far as I can see the paper from O'Neill group was submitted in summer last year while O'Neil et al submitted their manuscript just now. During the time between summer 2017 and now, a couple of other major works in the field of itaconate biology have been published. Authors need to familiarize themselves with the field developments that happened, specifically, two of the following works:

First, in the fall 2017, Vamsi Mootha's lab published the work (Shen et al, Cell 2017) that clearly shows that itaconate (in its natural form) penetrates into the cells directly, i.e. it is itself membrane-permeable metabolite. Second, recent work from Stallings and Diamond labs (Nair et al, J Exp Med, 2018) shows that itaconic acid directly (rather than any itaconate derivative) rescues transcriptional response of Irg1ko macrophages to in vitro MTb infection - also indicating that itaconate is a membrane permeable metabolite in its natural form.

So, as of March 2018, when talking about natural itaconate (TCA-cycle derived metabolite, as authors put it), there is no need to employ various chemical derivatives - itaconate itself can and SHOULD be used directly. Given the results from Mootha and Diamond labs, it is scientifically inappropriate to talk about itaconate in the manuscript and figure caption (even showing itaconate on the schematic panel Fig.4g) while presenting experimental data only on octyl-itaconate treatment. Hence, either authors have to reformulate the paper to explicitly state that octyl-itaconate inhibits STING pathway, or to show results with the actual physiological itaconate or Irg1ko to support the claims made in current write-up.

This is a valid point – which has been made by more than one reviewer. In accordance to what is suggested by both reviewers, we have changed the wording and the graphics to exclude any misunderstandings on what we are claiming. The term 4-OI is now used throughout the manuscript.

Reviewer #2

1. Targeting of Nrf2 in cancer cell lines significantly alters their biosynthesis and growth capacity, and this presumably translates to compromised viral growth as well. The claim that Nrf2 impairs innate immunity is therefore somewhat strong. Some additional data should be provided on cell growth to highlight this discrepancy and the language toned down. Alternatively, including controls that show how targeting STING levels independently impacts viral titer (or does not) would be helpful.

As suggested by the reviewer we have provided additional experimental data showing that in the cell lines we use (A549 and HaCat), STING levels independently impact HSV infection/replication. These data are now added to the manuscript and displayed in Fig. S12. Further, in Fig. S5, we now also provide novel data which demonstrate that overexpression of Nrf2 reduces known STING-dependent immune responses in an ISRE Luciferase HEK293 cell-based assay.

2. The authors routinely use the term itaconate when referring to the OI ester (particularly in subtitles). These molecules are different (and behave differently in cells as demonstrated by the Mills paper), so when using OI it should be referred to as such. The same applies to the drug DFO and its unesterified counterpart fumarate. Revision is necessary throughout the manuscript.

This is a valid observation and we have corrected this throughout the manuscript.

3. Along these lines, it would be informative to investigate how Irg1 KD or KO impacts TMEM173 mRNA stability over time.

A better characterization of the role of IRG1 is certainly relevant. However, in our hands the silencing of IRG1 leads to extensive cell death in the hMDMs making experiments very difficult to interpret IRG1 KO/KD experiments in our setup. The photo below shows differentiated hMDMs where we have silenced either, Nrf2 or IRG1. As indicated by the photo the cells die when IRG1 is silenced. Why human MDMs are that sensitive to silencing of IRG1 is highly interesting – but we feel that investigation of this phenomenon is too far outside the scope of this manuscript to pursue.

4. In Figure 4b the authors show that unstimulated PMDC05 cells have mM levels of itaconate in supernatant (unclear what “supernatant” of cells even is – medium?). This indicates that TLR4 signaling is strongly active even basally. Please clarify as most unstimulated cells do not produce so much itaconate and define supernatant. Furthermore, the fold change in levels should be orders of magnitude rather than 2-fold.

To more precisely indicate what has been measured we have altered the term “supernatant” to “cell medium”. We do not know why PMDC05 cells have high basal production of itaconate. The levels are in the mM range – which is roughly what has been observed by others (*Mills et al., 2018*). And the increase in itaconate following TLR engagement of 5-6mM is equivalent to increases that have been reported in other cell types. As an example, the increase observed by *Mills et al* was 4mM. Although a thorough investigation of what drives basal production of itaconate in this particular cell type and how this differs from other cell types is indeed interesting – we feel that it is outside the scope of this manuscript. We have, however, added more data to this section and have added measurements of cell pellets to support the findings we made in the supernatant (cell medium). These measurements supported an increase in itaconate in response to LPS (Fig. 4c).

Reviewer #3

1. Fig. 1c,d,e,f: What do the individual points show? Independent samples of an experiment or independent experiments? This is not totally clear from the figure legend.

We agree with the reviewer that this is not clear from the legend and we have changed the legend-text to make this clearer.

2. Fig. 1m,4q and S7. The protein levels of the partly Nrf2 regulated gene HO-1 (PMID: 17942419) from western blots are shown at the representative photos to be increased after siNrf2 treatment. It is understandable that its baseline levels are very low and this could be due to variability but please explain this based on all your results. It would be advisable to use other Nrf2 target genes as well for validation purposes such as Nqo1, Txnrd1 etc.

To address this concern, we blotted the hMDM samples for other known Nrf2 inducible genes (NQO-1 and p62/SQSTM1). Here, the expression of NQO-1 and p62/SQSTM1 is clearly repressed when Nrf2 is silenced. Thus, the slight increase in basal levels of HO-1 observed in the untreated samples when Nrf2 is silenced is – as suggested by the reviewer – a low baseline variability (now Fig. 1n and 4q).

3. Lines 180-182. The results from ChIP-Seq are very interesting and combined with the mRNA levels of STING point to the direction that indeed Nrf2 should affect mRNA stability. Authors should consider changing the expression from “...a novel post-translational...” to “... a novel post-transcriptional...”.

This was of course an error and has now been changed accordingly.

4. Lines 205-206. In general, it is a bit confusing jumping from one cell line to another during the experimental results. The authors should clearly state the advantages of using a specific cell line for a certain experiment like they did for example with A549. In this case it is not so evident why they picked the PMDC05 line.

This is a valid point. We have chosen the PMDC05 cell line as it is reported to express a wide variety of Toll-like receptors and to respond well to their ligands. We changed the text accordingly.

5. Fig. 4h. It is not acceptable to present in a final paper these results without enough n number to perform statistics.

In accordance with the reviewer's comments, we have performed the appropriate statistical tests for this experiment. This is now indicated in the figure and explained in the legend. The graph is now representative of 2 independent experiments each performed in triplicate.

6. The point that Nrf2 can be the mediator of STING repression by itaconate is novel and very interesting to the field and is proven by siRNA studies (mainly shown in Fig.4). As these results are central to this work, it would be advisable to use at least one more alternative method of silencing Nrf2 (e.g. Crispr as in the other experiments or another siRNA construct) to ensure that these results can be replicated and reduce the possibility of off target effects.

We agree with the reviewer's comment. To ensure that our results can be replicated and reduce the possibility of off-target effects when using the initial Nrf2 siRNA sequence (sc-37007 Santacruz), we performed additional experiments using a different human siRNA sequence from the same provider (sc-44332 Santacruz). As initially observed, silencing of Nrf2 using the new siRNA sequence also led to an increase in STING and a reduction in HO-1 expression in A549 cells. These data are now displayed in a new panel S2B. Furthermore, to strengthen the point that Nrf2 is a mediator of STING repression upon 4-octyl-itaconate treatment, HaCat cells were silenced for Nrf2 using both siRNA sequences and subsequently stimulated with 4-OI. Interestingly, both siNrf2 sequences prevented the repressive effect 4-OI had on STING. This new piece of data is now included in Fig. S14 and supports our claim that 4-OI mediates STING repression via Nrf2

7. Moreover, in line 268 it would be better to use the expression " is mostly regulated by the transcription factor Nrf2..." as silencing Nrf2 does not always lead to complete reversal of the endpoints such as TMEM173 expression (Fig. 4o,p).

As suggested we have toned down the wording here.

Reviewers' comments:

Reviewer #1 (Remarks to the Author):

The authors have added considerable amount of new experimental data that unequivocally demonstrate connection between Nrf2 and STING regulation. This part of the work is now strong and sufficiently substantiated, and definitely warrants publication. However, i still have an issue with the claims on metabolic rewiring. Specifically, the authors do not provide enough data to justify the title "Nrf2 is negative regulator of Sting DURING METABOLIC REWIRING".

I detail the specific reasons below, but in short i would suggest two potential routes to the authors - either softening the language about metabolic rewiring (for example Nrf2 is negative regulator of STING suggesting its role in metabolic rewiring") or keeping the same title and performing explicit experiments to prove the statements as they are currently stated by addressing the points below.

Furthermore, while I like the connection between Nrf2 and Sting a lot and think that this is novel, however, there seem to be certain lack in expertise in immunometabolism judging by sometime random citations and interpretations.

As such, I will take a moment to further educate the authors on the subject and hopefully it will help to improve the manuscript.

Here are important specific point:

1. First, on rows 84-85 authors state

Metabolic reprogramming is now known to include an increase in glycolysis and a two-point interruption of the tricarboxylic acid (TCA) cycle

and cite work by Mills et al (reference 23) which is completely irrelevant for this statement.

This work neither discovers glycolysis importance nor two-point interruption of the TCA cycle in activating macrophages.

The appropriate citations for this kind of statement would be:

for glycolysis - Everts et al. 2012. PMID 22786879

for breakpoints in TCA cycle - Jha et al. 2015. PMID 25786174 (for Idh breakpoint); and of course Tanahill et al, Cordes et al and Lampropoulou et al (for Sdh breakpoint).

2. Second, on rows 229-231 authors state:

LPS was long known to activate Nrf2 but more recently the O'Neill's group identified that accumulation of the citrate-derived metabolite itaconate bound Keap1 to promote Nrf2 activity in LPS-treated immune cells.

and cite recently published work by Mills et al (Nature 2018).

This statement in the context of this citation is actually exactly NOT TRUE.

The authors should re-read the paper by O'Neill group and recognize that in that work, Mills et al didn't observe any reactivity of actual itaconate with Keap1. All their data are about reactivity of octyl-itaconate - Keap1 binding detected by mass-spec is binding with octyl-itaconate, not with itaconate. In fact Mills et al show that itaconate and octyl-itaconate are not the same - their OI inhibits il1b already at concentration of 62uM, and yet supplementary Fig (Extended Data Figure 3e in their paper) clearly shows that OI does not produce any itaconate when put on macrophages at this concentration whether before or after LPS stimulation.

As such, it is simply incorrect to attribute the discovery of the itaconate electrophilicity to work by Mills et al. They did great job illustrating anti-inflammatory effect and electrophilicity of octyl-

itaconate, but not endogenous itaconate. The work that appeared concurrently - Bambouskova et al (Nature, 2018, PMID 29670287) has shown exactly what the authors are alluding to - that endogenous itaconate is indeed electrophile that activates Keap1-Nrf2 pathway and reacts with glutathione, as proved by experiments with Irg1ko vs WT macrophages (see Fig. 1 in Bambouskova et al). I believe that this would be most appropriate citation for the statement on rows 229-231.

3.

This discrepancy between octyl-itaconate and itaconate (that was hidden in supplementary material of Mills et al) is ultimately the reason why the authors should either soften their claims about the importance of metabolic rewiring or validate them with usage of actual itaconate or siIrg1 or Irg1ko or similar means.

For example, the metabolomic measurements of itaconate that you provide are at 24h, which show that increase in concentration of itaconate by ~5mM (Fig. 4b) does not actually lead to any changes in STING concentration at 24h despite the fact that there is well defined upregulation of HO-1 (left panel of Fig.4e).

4.

The other reason to worry about it is differences in concentration - increase of measured ITACONATE by about 5mM does not yield considerable effect on STING by 24h, while OI is effective at 50-100uM range, again indicating that the anti-inflammatory effect of OI is due to its chemical electrophilicity, not due to its hypothetical yield of physiological form of itaconate. If authors want to debate that they should explicitly measure yield of itaconate upon treatment with OI and compare it to the itaconate upregulation during LPS response.

5.

For all the reasons mentioned above, it also might be more accurate to call octyl-itaconate as cell-permeable DERIVATIVE of itaconate, as opposed to cell-permeable FORM of itaconate, since it does get inside the cells and works as octyl-itaconate inside the cells (not as itaconate) based on Mills et al data.

6.

Finally, CpG stimulation induces quite dramatic Irg1 upregulation and produces itaconate, as well as PIC (at least in mouse cells). Your figure 4e suggests that the potential effect of itaconate at 72h is only working in LPS and Gardi, while presumably (judging by the upregulation of the Irg1) there should equal amount of itaconate produced in each of those stimulations. The different levels of HO1 might be due to different extent of iNOS upregulation by different stimuli, which leads to oxidative stress and further Nrf2 activation. So, the connection itaconate-STING is very circumstantial here.

7.

I want to stress that I find the potential connection between metabolic rewiring and STING regulation very exciting and worth publishing even in a form of potential possibility. At present, the authors do not have enough data to definitively prove the existence of such connection, while there is definitely enough data to SUGGEST the existence of such connection. Given the novelty and completeness of the data on Nrf2-STING connection, I think the manuscript would be completely acceptable for publication even with the current amount of data provided that authors reformulate their current narrative regarding metabolic rewiring to soften the claims to indicate the potential nature of the connection and double check the accuracy of their citations.

Reviewer #2 (Remarks to the Author):

The authors have responded to my critiques. Amazingly, I'm already exhausted by the science of

itaconate!

They should note, there are other labs than O'Neill's that have measured itaconate (or more aptly measured that unstimulated cells do not produce it). If PMDC05 cells are basally producing mM levels of itaconate, how can they be considered "unstimulated?"

Reviewer #3 (Remarks to the Author):

The authors have successfully and convincingly responded to all my comments by changing the text and performing additional experiments where needed.